# MoAT: Multi-Modal Augmented Time Series Forecasting

## Abstract

Time series forecasting plays a pivotal role in various domains, facilitating optimized resource allocation and strategic decision-making. However, the scarcity of training samples often hinders the accuracy of the forecasting task. To address this, we explore the potential of leveraging information from different modalities that are commonly associated with time series data. In this paper, we introduce MoAT, a novel multi-modal augmented time series forecasting approach that strategically integrates both feature-wise and sample-wise augmentation methods to enrich multi-modal representation learning. It further enhances prediction accuracy through joint trend-seasonal decomposition across all modalities and fuses the information for the final prediction. Extensive experiments show that MoAT outperforms state-of-the-art methods, resulting in a substantial reduction in mean squared error ranging from 6.5% to 71.7%, which demonstrates the effectiveness and robustness in addressing the limitations imposed by data scarcity. The datasets and code are available at `https://anonymous.4open.science/r/MoAT-201E`.

## 1 Introduction

Time series forecasting is a fundamental task with widespread applications across various domains, including finance, healthcare, energy management, and environmental monitoring (Sawhney et al., 2020; Chou & Tran, 2018; Ong et al., 2016). Accurate time series forecasting is essential for making informed decisions, optimizing resource allocation, and enhancing strategic planning. Notable progress has been achieved by various time series forecasting models (Wu et al., 2021; Zhou et al., 2022b; Nie et al., 2022; Zhang & Yan, 2022) following recent advancements in deep learning. However, in numerous scenarios, time-series data suffer from a scarcity of training samples, posing significant challenges in predicting future trends (Zhou et al., 2022c). Recall the COVID-19 outbreak, where accurate predictions of confirmed cases, fatalities, and recoveries were crucial in public health responses and resource allocation. Nevertheless, due to the limited amount of data available at the time, it was extremely challenging to make reliable predictions about future epidemic occurrences solely from time-series data, despite its critical importance (Gothai et al., 2021).

To address this limitation, leveraging the information from other modalities that associated with time series data emerges as a potential solution. For example, within the healthcare domain, clinical notes have been used to enhance the accuracy of predicting patient mortality (Deznabi et al., 2021; Yang et al., 2021). In the financial domain, leveraging text data from social media has proven to be effective in enhancing the accuracy of predicting individual stock movements (Du & Tanaka-Ishii, 2020; 2022; Sawhney et al., 2020). Another motivative example is evident in Fig. 1 which provides a visual representation of how distinct data modalities can shed light on similar dynamics. We can observe two prominent drops in the time series data marked as 1 and 2 in Fig. 1a, which correspond to significant changes in the similarity of text embeddings depicted in Fig. 1b. Expanding the scope of the training dataset becomes possible when taking into account both data sources, as shown in a 2D space in Fig. 1c. These findings suggest the potential of using text data to enhance the precision of time series forecasting.

The aforementioned studies that aid the time-series modeling by exploiting knowledge from other modalities, mostly from text data, can be formulated as data augmentation which is widely used to improve the model performance by enlarging the distribution covered by training dataset (Qin et al.,

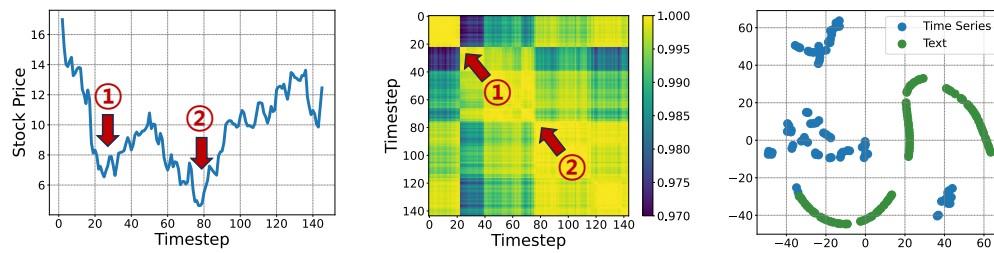

(a) Visualization of stock price time series data

(b) Similarity heatmap of financial news embedding

(c) Visualization of the augmented training data

Figure 1: (a) Visualization of the Stock-Index dataset's time series data used in this study; (b) Corresponding similarity heatmap of financial news embedding, and (c) Visualization of the augmented training dataset. In the time series plot (a), there are two noticeable drops indicated by circled 1 and 2, which coincide with significant changes in text embedding similarity shown in (b). While these two data modalities are synchronized in time, they offer distinct information, as illustrated in a 2D space using t-SNE (c), thereby expanding the coverage of the training dataset.

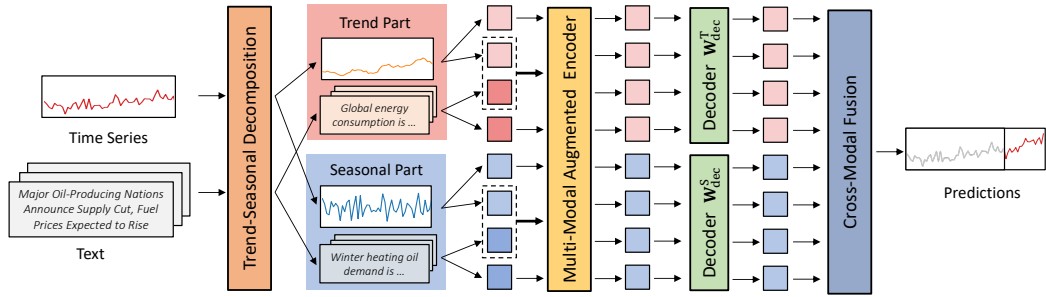

Figure 2: The architecture of MoAT. Time series and text data are jointly decomposed into trend and seasonal components. These components are fed to the multi-modal augmented encoder which incorporates both sample-wise and feature-wise multi-modal data augmentation. The resulting outputs are decoded using component-specific decoders. Finally, the predictions for trend and seasonality are combined through a cross-modal fusion scheme to generate the final predictions.

2020; Zhou et al., 2022a; Baltrušaitis et al., 2018). Data augmentation can be approached from two distinct angles, each offering unique benefits. The first method involves sample-wise augmentation, treating samples originating from different modalities as additional training examples (Xing et al., 2019; Pahde et al., 2021; Lin et al., 2023). The second method integrates supplementary information from diverse modalities into each individual sample (Sun et al., 2019; Guo et al., 2020; Zheng et al., 2021; Shi et al., 2021). Each approach improves model performance and robustness in its distinct way. This motivates us to strategically integrate both augmentation techniques, allowing us to leverage the unique advantages offered by both augmentation approaches.

Notably, the aforementioned studies utilizing text data for time series tasks have primarily concentrated on classification tasks (e.g., predicting whether a patient will survive or not, or the trend of stock movements). In this work, we present MoAT, a general multi-modal augmented framework designed to enhance forecasting accuracy by integrating information from various sources. Illustrated in Fig. 2, our approach comprises three main components: multi-modal augmentation, cross-modal fusion, and trend-seasonal decomposition. The augmentation encoder integrates both sample-wise and feature-wise multi-modal data augmentation techniques to enhance the quality and diversity of the training data. The resulting outputs are decoded using component-specific decoders. MoAT then facilitates the fusion of decoded representations from different modalities, enabling a more comprehensive understanding of the underlying processes governing time series behavior. To further enhance its capabilities, MoAT incorporates a joint trend-seasonal decomposition module, which generates distinct embedding to capture trend and seasonal dynamics from both time series and text information. Through extensive experiments, MoAT demonstrates substantial improvements, achieving a 6.5%-71.7% reduction in mean square error (MSE). In summary, this paper makes the following contributions:

• We introduce MoAT, a multi-modal augmented time series forecasting framework that addresses data scarcity issues by employing both sample-wise and feature-wise multi-modal augmentation and cross-modal fusion to enhance training data quality and diversity.

- MoAT further enhances forecasting accuracy through joint trend-seasonal decomposition module, which extracts distinct embedding to capture trend and seasonal patterns from both time series and text information.

- Extensive experiments on six multi-modal datasets confirm the effectiveness and robustness of our proposed model. Furthermore, we release all compiled multi-modal time series datasets[1] to facilitate further research in this domain.

## 2 RELATED WORK

**Time series forecasting.** Given its significant importance, time series forecasting has been extensively researched over a long period. In its early stages, classical deep learning methods were commonly applied. For example, TCN (Lea et al., 2017) uses CNN, and DeepAR (Salinas et al., 2020) utilizes RNN to capture temporal dependencies within time series. In addition, LSTnet (Lai et al., 2018) integrates both CNN and RNN in its approach. Recently, inspired by the success of Transformers (Vaswani et al., 2017) in various domains, including natural language processing (NLP) (Devlin et al., 2019), computer vision (CV) (Dosovitskiy et al., 2020), and speech processing (Dong et al., 2018), they have been actively used for time series forecasting. Informer (Zhou et al., 2021) is based on an efficient ProbSparse self-attention for capturing cross-time dependencies. Autoformer (Wu et al., 2021) uses time series decomposition and auto-correlation to forecast time series. FEDformer (Zhou et al., 2022b) uses Fourier or wavelet transforms to apply attention in the frequency domain. Despite these advancements, comparative evaluations have shown that these methods can be outperformed by DLinear (Zeng et al., 2023), a very simple linear model, shedding light on the limitations of Transformer-based time series forecasting models. However, this claim is contradicted by recent patch-level (or segment-level) Transformer models. Inspired by the effectiveness of patching in NLP (Devlin et al., 2019) and CV (Dosovitskiy et al., 2020; Bao et al., 2021), PatchTST (Nie et al., 2022) and Crossformer (Zhang & Yan, 2022) employ time series patches as the input for the Transformer encoder. Utilizing time series patching allows the model to capture temporal dependencies beyond individual data points, facilitating a comprehensive understanding of time series patterns and leading to more accurate forecasting.

**Multi-modal data augmentation.** Data augmentation is widely used to improve the model performance by enlarging the distribution covered by training data. An extensive literature on uni-modal data augmentation exists across diverse domains including images (Shorten & Khoshgoftaar, 2019), texts (Feng et al., 2021), and time series Wen et al. (2021). Here, we summarize the augmentation methods applied to multi-modal datasets, which aggregate data from multiple distinct sources. One approach involves incorporating additional training samples from different modalities, which is especially beneficial when the specific modality of interest has limited available data. This can be achieved by simply using samples from other modalities as extra training data (Lin et al., 2023; Pahde et al., 2021), or by generating synthetic training samples through transformations from multiple sources (Hao et al., 2023). An alternative and orthogonal approach is to enrich the information within each sample by integrating additional data from different modalities. A simple and direct method is to concatenate features from multiple modalities and feed them collectively into the Transformer encoder, allowing for cross-modal learning (Sun et al., 2019; Zheng et al., 2021; Shi et al., 2021). In summary,

- **Sample-wise augmentation** in multi-modal data treats samples originating from different modalities as additional samples to train the model (Xing et al., 2019; Pahde et al., 2021; Lin et al., 2023). This approach is particularly advantageous when different modalities contribute complementary information, each capturing unique aspects of the underlying concept. For example, incorporating samples from text data can assist in clarifying the decision boundary of the image classifier.

- **Feature-wise augmentation** integrates additional information from different modalities into each sample (Sun et al., 2019; Guo et al., 2020; Zheng et al., 2021; Shi et al., 2021). Enriching each sample by incorporating features from diverse sources enables the model to understand complex relationships and patterns within the data, leading to more accurate predictions or classifications. For example, concatenating text features with image features can assist in adjusting the weights of the image classifier.

---

[1]`https://anonymous.4open.science/r/MoAT-201E`

**Multi-modal for time series.** Additional information, mostly in textual data, has been leveraged to solve time series tasks, primarily within specific domains. For example, within the healthcare domain, clinical notes have been used to enhance the accuracy of predicting patient mortality (Deznabi et al., 2021; Yang et al., 2021) or to optimize the management of patients' ICU stays (Khadanga et al., 2019). In the financial domain, leveraging text data from social media has proven to be effective in enhancing the accuracy of predicting individual stock movements (Du & Tanaka-Ishii, 2020; 2022; Sawhney et al., 2020). Importantly, previous studies that utilize text data for time series tasks have primarily concentrated on classification tasks (e.g., predicting whether a patient will survive or not, or whether the stock price will increase or decrease the following day), differing from our specific focus on forecasting. In this work, we propose MoAT, a general time series forecasting model that integrates text information to enhance prediction accuracy.

## 3 PROPOSED METHOD

We propose MoAT, a multi-modal time series forecasting model that leverages information from different modalities to improve the accuracy of the forecasting task, as illustrated in Fig. 2.

### 3.1 PROBLEM STATEMENT

The problem of multivariate time series forecasting is defined as predicting the future $T$ steps of time-series values $\mathbf{X} = (x_{L+1}, ..., x_{L+T}) \in \mathbb{R}^{T \times C}$, based on the past $L$ steps of time series data $\mathbf{Y} = (x_1, ..., x_L) \in \mathbb{R}^{L \times C}$, where $C$ represents the dimension or the number of channels of the time series data. At each historical time step $t = 1, ..., L$, there is a set $\mathcal{D}_t = \{\mathcal{D}_{t,1}, ..., \mathcal{D}_{t,|\mathcal{D}_t|}\}$ of data available from other modalities, which has the potential to improve forecasting accuracy.

### 3.2 MoAT: MULTI-MODAL AUGMENTED TIME SERIES FORECASTING

Here, we describe our framework for time-series forecasting using text data as an example. This approach can readily be expanded to include other modalities such as image and audio data.

**Patch-wise embedding.** Beyond point-wise, analyzing time series data at the segment level provides richer insights for understanding its underlying dynamic patterns. For example, when examining stock price movements, it becomes crucial to analyze how prices change over time, rather than focusing on the individual price on a specific day. The effectiveness of patching time series data has been well-demonstrated, particularly in transformer-based forecasting models (Nie et al., 2022; Zhang & Yan, 2022), and thus MoAT also adopts this approach. In our approach, we represent both time series and text as an identical number of patches, as described below.

To patch **time series** $\boldsymbol{x}^{(i)} = (x_1^{(i)}, ..., x_L^{(i)}) \in \mathbb{R}^L$ of the $i$-th channel, we segment it into multiple (non-)overlapping patches. Precisely, given the patch length $P$ and the stride $S$, we segment $\boldsymbol{x}^{(i)}$ into $N$ patches, each of length $P$, denoted as $\mathbf{p}^{(i)} = (\mathbf{p}_1^{(i)}, ..., \mathbf{p}_N^{(i)}) \in \mathbb{R}^{N \times P}$ where two consecutive patches share $S$ values. Here, the number $N$ of patches is $\lfloor \frac{L-P}{S} \rfloor + 2$ if we pad the last value of the time series $S$ times before patching. Then, we map these patches into a $d$-dimensional latent space using a learnable linear projection $\mathbf{W}_{\text{time}} \in \mathbb{R}^{P \times d}$, i.e., $\mathbf{z}_{\text{time}}^{(i)} = \mathbf{p}^{(i)} \mathbf{W}_{\text{time}} \in \mathbb{R}^{N \times d}$.

To patch **text data**, we require a different approach due to the variability in the number of texts and their unordered nature at each timestep. Specifically, we are given a sequence $\mathcal{D} = (\mathcal{D}_1, ..., \mathcal{D}_L)$ of sets of documents, where at each timestep $t$, we have a set $\mathcal{D}_t = \{\mathcal{D}_{t,1}, ..., \mathcal{D}_{t,|\mathcal{D}_t|}\}$ of an arbitrary number of texts. Firstly, we use the pretrained language model (denoted as "PLM" below) to represent each text $\mathcal{D}_{t,j}$ as a $d'$-dimensional embedding vector, i.e., $\boldsymbol{d}_{t,j} = \text{PLM}(\mathcal{D}_{t,j}) \in \mathbb{R}^{d'}$. This yields an embedding matrix $\boldsymbol{d}_t = [\boldsymbol{d}_{t,1}, ..., \boldsymbol{d}_{t,|\mathcal{D}_t|}] \in \mathbb{R}^{|\mathcal{D}_t| \times d'}$ of texts at each timestep $t$.

Following the segmentation approach similar to that of the time series, we segment text data into multiple patches, each spanning $P$ timesteps. To represent a text patch covering the time span from $t_a$ to $t_b$, where $t_b - t_a + 1 = P$, our goal is to aggregate text embeddings within this interval into a single embedding. To this end, we collect text embeddings spanning from $t_a$ to $t_b$, $\boldsymbol{d}_{t_a:t_b} = \boldsymbol{d}_{t_a} \oplus \cdots \oplus \boldsymbol{d}_{t_b} \in \mathbb{R}^{|\mathcal{D}_{t_a:t_b}| \times d'}$, where $\oplus$ is the vertical concatenation operation. Since various documents may hold different degrees of significance, we apply attentive pooling to aggregate their

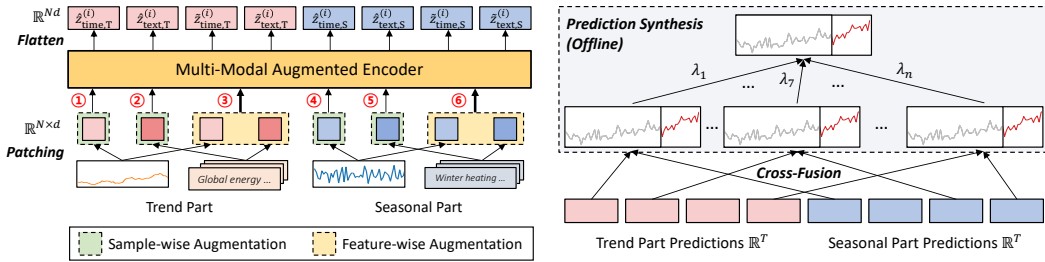

(a) Multi-modal augmented encoder (b) Cross-modal fusion and offline synthesis

Figure 3: Key components of MoAT. (a) The multi-modal augmented encoder is utilized across six distinct input patches, labeled 1 to 6, which are obtained by augmenting multi-modal inputs either in a sample-wise or feature-wise manner. Note that inputs 1 to 6 are processed independently with the shared encoder. (b) Using the eight different representations derived from trend and seasonal components, MoAT generates multiple predictions through cross-fusion. These predictions are then aggregated to generate the final prediction using an offline synthesis approach.

embeddings while assigning varying weights as follows:

$$\text{Softmax}\left(\tanh(\boldsymbol{d}_{t_a:t_b}\mathbf{W}^{(i)} + \boldsymbol{b}^{(i)})\mathbf{V}^{(i)}\right)\boldsymbol{d}_{t_a:t_b} \in \mathbb{R}^{d'} \tag{1}$$

where $\mathbf{W}^{(i)} \in \mathbb{R}^{d' \times d}$, $\boldsymbol{b}^{(i)} \in \mathbb{R}^{d}$, and $\mathbf{V}^{(i)} \in \mathbb{R}^{d \times d}$ are learnable weights specific to the $i$-th channel. It is important to note that we utilize channel-specific parameters when applying attention, which enables us to generate text patches that are unique to each channel. Consider financial markets as an example, where the importance and relevance of documents can vary for different stocks or companies. After mapping the patches into a $d$-dimensional latent space using a trainable linear projection $\mathbf{W}_{\text{text}} \in \mathbb{R}^{d' \times d}$, we obtain embeddings of text patches $\mathbf{z}_{\text{text}}^{(i)} \in \mathbb{R}^{N \times d}$ of the $i$-th channel. These patch embeddings temporally align with the time series patches, $\mathbf{z}_{\text{time}}^{(i)}$.

**Multi-modal augmented encoder.** Now that we have obtained patch embeddings, $\mathbf{z}_{\text{time}}^{(i)}$ for time series and $\mathbf{z}_{\text{text}}^{(i)}$ for text data, in the $i$-th channel, we proceed to feed them into the shared encoder illustrated in Fig. 3a. In this step, we leverage multi-modal data augmentation strategies from two distinct angles, each providing unique benefits, all with a common goal of forecasting time series.

Commonly, we introduce modality-specific learnable positional embeddings $\mathbf{W}_{\text{time}}^{\text{pos}} \in \mathbb{R}^{N \times d}$ and $\mathbf{W}_{\text{text}}^{\text{pos}} \in \mathbb{R}^{N \times d}$ for time series and text, respectively. In addition, we employ a vanilla Transformer encoder (denoted as "Encoder" below) equipped with multi-head attention. The encoder yields patch representations equivalent to the number of input patches, and we assume that these representations are then flattened into a single embedding vector.

To employ the **sample-wise augmentation**, we use embeddings from the two modalities as independent training samples. Specifically, we pass them separately to the shared encoder, preventing any explicit interactions between patches across modalities:

$$\hat{\mathbf{z}}_{\text{time}}^{(i)} = \text{Encoder}\left(\mathbf{z}_{\text{time}}^{(i)} + \mathbf{W}_{\text{time}}^{\text{pos}}\right) \in \mathbb{R}^{Nd} \quad \text{and} \quad \hat{\mathbf{z}}_{\text{text}}^{(i)} = \text{Encoder}\left(\mathbf{z}_{\text{text}}^{(i)} + \mathbf{W}_{\text{text}}^{\text{pos}}\right) \in \mathbb{R}^{Nd} \tag{2}$$

As a result, the encoder processes twice the number of training samples, enriching diversity within the training data, which potentially improves generalization and robustness in the model. We refer to these embeddings as *single-modal* representations as they originate from individual modalities.

To employ the **feature-wise augmentation**, we feed the encoder with the concatenated embeddings derived from time series and text, allowing cross-modal interactions across modalities:

$$\hat{\mathbf{z}}_{\text{joint}}^{(i)} = \text{Encoder}\left(\mathbf{z}_{\text{time}}^{(i)} + \mathbf{W}_{\text{time}}^{\text{pos}} \,\|\, \mathbf{z}_{\text{text}}^{(i)} + \mathbf{W}_{\text{text}}^{\text{pos}}\right) \in \mathbb{R}^{2Nd} \tag{3}$$

where $\|$ is the concatenation operator. Note that, since the input comprises of $2N$ patches, the output $\hat{\mathbf{z}}_{\text{joint}}^{(i)}$ is the concatenation of $2N$ patches. We subsequently split it into two halves to obtain $\tilde{\mathbf{z}}_{\text{time}}^{(i)} \in \mathbb{R}^{Nd}$ and $\tilde{\mathbf{z}}_{\text{text}}^{(i)} \in \mathbb{R}^{Nd}$, i.e., $\hat{\mathbf{z}}_{\text{joint}}^{(i)} = \tilde{\mathbf{z}}_{\text{time}}^{(i)} \,\|\, \tilde{\mathbf{z}}_{\text{text}}^{(i)}$. We refer to these embeddings as *cross-modal* representations as they result from joint interactions across patches from different modalities.

***Remarks.*** Our multi-modal augmented encoder yields two distinct types of representations, single-modal (i.e., modality-specific) and cross-modal (i.e., modality-fused), each contributing unique information. While most existing works tend to utilize either one of these aspects (Lin et al., 2023;

Shi et al., 2021), MoAT leverages both perspectives of the multi-modal data, which strategically broadens its learning spectrum. In addition, the usefulness of these representations may vary across datasets, and in the subsequent descriptions, we describe how MoAT adaptively utilizes them to achieve accurate time series forecasting.

**Joint trend-seasonal decomposition.** This technique is one of the most valuable techniques for improving the predictability of raw time series data (Cleveland et al., 1990; Hamilton, 2020). Notably, recent forecasting models have incorporated time series decomposition into their frameworks due to its effectiveness (Wu et al., 2021; Zhou et al., 2022b; Zeng et al., 2023). Specifically, the seasonal-trend decomposition employs a moving average kernel on the input time series to extract the *trend-cyclic* component of the data. Then, the *seasonal* component is derived as the difference between the original input time series and the extracted trend component. Inspired by its effectiveness, we enhance the method by employing time series decomposition.

To decompose **time series**, we use the seasonal-trend decomposition to yield trend part $x_{\mathrm{T}}^{(i)}$ and seasonal part $x_{\mathrm{S}}^{(i)}$, each of which explain different aspects of the time series. From each component, we compute patch embeddings $\mathbf{z}_{\text{time,T}}^{(i)}$ and $\mathbf{z}_{\text{time,S}}^{(i)}$ using the aforementioned approach.

To decompose **text data**, we employ two distinct and independent sets of attention parameters, $\theta_{\mathrm{T}}^{(i)} = \{\mathbf{W}_{\mathrm{T}}^{(i)}, \boldsymbol{b}_{\mathrm{T}}^{(i)}, \mathbf{V}_{\mathrm{T}}^{(i)}\}$ and $\theta_{\mathrm{S}}^{(i)} = \{\mathbf{W}_{\mathrm{S}}^{(i)}, \boldsymbol{b}_{\mathrm{S}}^{(i)}, \mathbf{V}_{\mathrm{S}}^{(i)}\}$ in Eq. 6 to obtain aggregated text embeddings for trend and seasonal components, respectively. This allows us to derive distinguished embeddings for capturing trend and seasonal dynamics from text information, highlighting documents differently. From each set of text embeddings, we derive two distinct sets of text patches, $\mathbf{z}_{\text{text,T}}^{(i)}$ and $\mathbf{z}_{\text{text,S}}^{(i)}$, representing trend-related and seasonal-related text information, respectively.

Once we obtain two sets of patch embeddings, $\mathbf{z}_{\text{time,M}}^{(i)}$ and $\mathbf{z}_{\text{text,M}}^{(i)}$, for each component $\mathrm{M} \in \{\mathrm{T}, \mathrm{S}\}$, we utilize the shared multi-modal augmented encoder described above. As shown in Fig. 3a, we feed time series and text patches from each component independently into the shared encoder, resulting in the following two sets, $\mathbf{Z}_{\mathrm{T}}^{(i)}$ and $\mathbf{Z}_{\mathrm{S}}^{(i)}$, of representations:

$$\mathbf{Z}_{\mathrm{T}}^{(i)} = \{\hat{\mathbf{z}}_{\text{time,T}}^{(i)}, \hat{\mathbf{z}}_{\text{text,T}}^{(i)}, \tilde{\mathbf{z}}_{\text{time,T}}^{(i)}, \tilde{\mathbf{z}}_{\text{text,T}}^{(i)}\} \quad \text{and} \quad \mathbf{Z}_{\mathrm{S}}^{(i)} = \{\hat{\mathbf{z}}_{\text{time,S}}^{(i)}, \hat{\mathbf{z}}_{\text{text,S}}^{(i)}, \tilde{\mathbf{z}}_{\text{time,S}}^{(i)}, \tilde{\mathbf{z}}_{\text{text,S}}^{(i)}\} \quad (4)$$

These representations originate from either single-modal or cross-modal inputs, granting us four unique information for both trend and seasonal components. This notably enriches the variety of training samples that the decoder can utilize to predict future time series.

**Multi-modal cross fusion.** Since we have decomposed the data into its trend and seasonal components, the next step involves recombining them to generate predictions. While a straightforward approach would be combining corresponding representations from $\mathbf{Z}_{\mathrm{T}}^{(i)}$ and $\mathbf{Z}_{\mathrm{S}}^{(i)}$ (e.g., combining $\hat{\mathbf{z}}_{\text{time,T}}^{(i)}$ and $\hat{\mathbf{z}}_{\text{time,S}}^{(i)}$ or $\tilde{\mathbf{z}}_{\text{text,T}}^{(i)}$ and $\tilde{\mathbf{z}}_{\text{text,S}}^{(i)}$) to yield four predictions, we propose to use a richer strategy. Instead of limiting the combinations to the corresponding pairs, we propose to aggregate all possible combinations of representations from $\mathbf{Z}_{\mathrm{T}}^{(i)}$ and $\mathbf{Z}_{\mathrm{S}}^{(i)}$ as shown in Fig. 3b. Thus, we minimize:

$$\mathcal{L}^{(i)} = \sum_{\mathbf{z}_{\mathrm{T}}^{(i)} \in \mathbf{Z}_{\mathrm{T}}^{(i)}} \sum_{\mathbf{z}_{\mathrm{S}}^{(i)} \in \mathbf{Z}_{\mathrm{S}}^{(i)}} \left\| \mathbf{z}_{\mathrm{T}}^{(i)} \mathbf{W}_{\text{dec}}^{\mathrm{T}} + \mathbf{z}_{\mathrm{S}}^{(i)} \mathbf{W}_{\text{dec}}^{\mathrm{S}} - \mathbf{Y}^{(i)} \right\|_2^2 \quad (5)$$

where $\mathbf{W}_{\text{dec}}^{\mathrm{T}} \in \mathbb{R}^{Nd \times T}$ and $\mathbf{W}_{\text{dec}}^{\mathrm{S}} \in \mathbb{R}^{Nd \times T}$ are the single-layer linear decoders for the trend and seasonal representations, respectively. The overall loss $\mathcal{L}$ is computed by the average of the $C$ individual losses from the $C$ channels, i.e., $\mathcal{L} = \frac{1}{C} \left( \mathcal{L}^{(1)} + ... + \mathcal{L}^{(C)} \right)$. This approach yields a total of 16 different combinations, each contributing to a distinguished prediction. Predicting from a wider range of representation combinations allows us to unravel a comprehensive understanding of the interplay between the trend and seasonal components.

***Discussion 1: How is multi-modality fused?*** To improve forecasting accuracy by integrating various modalities, it is essential to effectively fuse the information acquired from these sources. Initially, an *implicit fusion* of time series and text occurs in Eq. 3 by jointly passing patches from both modalities to the shared Transformer encoder, enabling cross-modal interactions among them. Cross-fusion, on the other hand, is an *explicit fusion* approach aimed at facilitating interactions between these modalities. By systematically combining all feasible pairs from $\mathbf{Z}_{\mathrm{T}}^{(i)}$ and $\mathbf{Z}_{\mathrm{S}}^{(i)}$ to make predictions, this approach explicitly considers all potential cross-modal fusions.

Table 1: Time series forecasting accuracy (in terms of MSE) in six real-world datasets. With effective integration of text information, MoAT provides the most accurate predictions for future time series. The best results are in **bold** and the second best are underlined. The average improvements (%) and ranks are reported on the right.

| Method | Fuel | Metal | Bitcoin | Stock-Index | Covid | Stock-Tech | Improv. | Rank |
|---|---|---|---|---|---|---|---|---|
| LightTS | $0.1930 \pm 0.0828$ | $0.1829 \pm 0.2065$ | $0.3109 \pm 0.2528$ | $1.6765 \pm 0.6564$ | $0.4504 \pm 0.1746$ | $1.0780 \pm 1.3709$ | 71.67 | 9.66 |
| DLinear | $0.1916 \pm 0.0346$ | $0.0757 \pm 0.0462$ | $0.0822 \pm 0.0213$ | $0.8250 \pm 0.0542$ | $0.2513 \pm 0.1221$ | $0.3546 \pm 0.1006$ | 43.81 | 7.33 |
| Autoformer | $0.1599 \pm 0.0266$ | $0.0529 \pm 0.0084$ | $0.0757 \pm 0.0038$ | $\mathbf{0.7856} \pm \mathbf{0.0408}$ | $0.2597 \pm 0.0213$ | $0.2115 \pm 0.0314$ | 34.92 | 6.00 |
| FEDformer | $0.1088 \pm 0.0151$ | $0.0488 \pm 0.0041$ | $0.0743 \pm 0.0068$ | $0.8004 \pm 0.0375$ | $0.2366 \pm 0.0179$ | $0.1369 \pm 0.0074$ | 24.22 | 4.50 |
| Pyraformer | $0.1366 \pm 0.0289$ | $0.2494 \pm 0.0590$ | $1.3467 \pm 0.1913$ | $0.9558 \pm 0.3773$ | $1.5176 \pm 0.2595$ | $11.0625 \pm 1.5610$ | 71.46 | 9.83 |
| Crossformer | $0.0962 \pm 0.0049$ | $0.1110 \pm 0.0344$ | $0.4256 \pm 0.0398$ | $1.5374 \pm 0.3410$ | $0.5700 \pm 0.0912$ | $7.0709 \pm 1.0786$ | 65.92 | 8.83 |
| PatchTST | $0.0913 \pm 0.0055$ | $\underline{0.0278} \pm \underline{0.0005}$ | $0.0527 \pm 0.0017$ | $0.8612 \pm 0.0315$ | $\underline{0.1774} \pm \underline{0.0077}$ | $0.1215 \pm 0.0026$ | 6.46 | 3.33 |
| MM-Linear | $0.1127 \pm 0.0212$ | $0.0504 \pm 0.0045$ | $0.0525 \pm 0.0050$ | $1.5321 \pm 0.3410$ | $0.1906 \pm 0.0129$ | $0.2680 \pm 0.1163$ | 32.48 | 5.66 |
| MM-LSTM | $0.1947 \pm 0.0014$ | $0.0607 \pm 0.0004$ | $0.0628 \pm 0.0014$ | $0.8571 \pm 0.0252$ | $0.1943 \pm 0.0088$ | $0.1665 \pm 0.0017$ | 30.45 | 6.33 |
| MM-TST | $\underline{0.0902} \pm \underline{0.0023}$ | $0.0285 \pm 0.0009$ | $\underline{0.0522} \pm \underline{0.0018}$ | $0.8919 \pm 0.0322$ | $0.1804 \pm 0.0162$ | $\underline{0.1201} \pm \underline{0.0025}$ | 6.65 | 3.16 |
| **MoAT** | $\mathbf{0.0816} \pm \mathbf{0.0016}$ | $\mathbf{0.0257} \pm \mathbf{0.0004}$ | $\mathbf{0.0494} \pm \mathbf{0.0006}$ | $\underline{0.8134} \pm \underline{0.0959}$ | $\mathbf{0.1727} \pm \mathbf{0.0020}$ | $\mathbf{0.1176} \pm \mathbf{0.0019}$ | - | **1.33** |

***Discussion 2: Why independent predictions?*** In our loss function (Eq. 5), we compute the loss for each of the 16 independent predictions, which are then summed to compute the final loss $\mathcal{L}^{(i)}$. An intuitively different approach would be to aggregate the representations themselves and forecast time series as a united prediction. However, our choice of the objective exploits a larger number of training samples, enabling the model to effectively capture a diverse set of input patterns. Information from different modalities may provide complementary insights (Xing et al., 2019; Mu et al., 2020; Wortsman et al., 2022), and thus utilizing each prediction independently could lead to a more comprehensive understanding of the underlying dynamics.

**Prediction synthesis.** In the inference phase, our primary objective is to generate a singular, definitive prediction from a set of multiple predictions generated by MoAT. While simply taking the mean of the 16 predictions appears as a straightforward approach, the reliability and accuracy of each prediction can vary across datasets, depending on the quality of the input modalities. Thus, instead, we implement a simple offline linear aggregation module consisting of 16 weight parameters $\lambda_1, \cdots, \lambda_{16}$, along with an intercept $b$, that assign varying weights to the predictions, as shown in Fig. 3b. Note that this synthesis module is trained offline, separately from the main module (e.g., encoder and decoder), and thus gradients are not shared.

***Discussion 3: Why offline synthesis?*** One might hypothesize that jointly learning $\lambda_1, \cdots \lambda_{16}$ along with the main module in an end-to-end manner could yield better results. However, in such a case, the model serves two roles: (1) accurately forecasting time series and (2) synthesizing predictions by learning their weights based on their importance. We disentangle the model's two functionalities by training the synthesis module offline. This allows MoAT to dedicate itself to generating accurate forecasts, while the offline module learns the weights associated with the generated predictions. The disentanglement of functionalities has been proven to enhance the modularity and efficiency of multi-modal models (Liang et al., 2022).

## 4 EXPERIMENTS

In this section, we evaluate MoAT by answering the following questions: **Q1. Accuracy:** Does MoAT accurately forecast time series? **Q2. Effectiveness:** Does each component of MoAT improve the performance? and **Q3. Versatility:** Does MoAT perform well in other settings as well? Supplementary results can be found in Appendix D.

### 4.1 SETTINGS

We first share the experimental settings where the experiments are conducted.

**Datasets.** We evaluate the performance of MoAT across six multi-modal datasets: Fuel, Gold, Stock-Index, Stock-Tech, Bitcoin, and Covid. These datasets, which will be made publicly available, consist of multivariate time series data ranging from daily to monthly resolution. Additionally, each dataset is accompanied by a collection of documents associated with each timestep. For more detailed descriptions of each dataset (e.g., statistics and preprocessing steps), refer to Appendix B.1.

**Baselines.** To evaluate our proposed method, we compare it with both uni-modal and multi-modal time series forecasting methods. As for uni-modal baselines, we consider linear models,

Table 2: The effects of (1) incorporating multi-modal data, (2) employing dual multi-modal augmentation schemes, and (3) considering other design choices discussed in Section 3. All components contribute collectively to improve the performance of MoAT.

| Method | Fuel | Metal | Bitcoin | Stock-Index | Covid | Stock-Tech | Improv. (%) |
|---|---|---|---|---|---|---|---|
| $\text{MoAT}_{\text{time}}$ | **0.0802** $\pm$ **0.0015** | 0.0264 $\pm$ 0.0007 | 0.0501 $\pm$ 0.0005 | 1.1879 $\pm$ 0.0354 | 0.1763 $\pm$ 0.0032 | 0.1223 $\pm$ 0.0018 | 6.613 |
| $\text{MoAT}_{\text{text}}$ | 0.1044 $\pm$ 0.0003 | 0.0310 $\pm$ 0.0002 | 0.0516 $\pm$ 0.0001 | 1.1750 $\pm$ 0.0293 | 0.1806 $\pm$ 0.0004 | 0.1501 $\pm$ 0.0010 | 16.696 |
| $\text{MoAT}_{\text{sample}}$ | 0.0814 $\pm$ 0.0016 | **0.0256** $\pm$ **0.0008** | **0.0491** $\pm$ **0.0005** | 0.8405 $\pm$ 0.0269 | 0.1748 $\pm$ 0.0026 | 0.1191 $\pm$ 0.0023 | 0.724 |
| $\text{MoAT}_{\text{feature}}$ | 0.0819 $\pm$ 0.0018 | 0.0262 $\pm$ 0.0013 | 0.0496 $\pm$ 0.0007 | 0.8383 $\pm$ 0.0302 | 0.1756 $\pm$ 0.0031 | 0.1200 $\pm$ 0.0035 | 1.532 |
| $\text{MoAT}_{\text{w.o.JD}}$ | 0.0823 $\pm$ 0.0013 | 0.0261 $\pm$ 0.0010 | 0.0501 $\pm$ 0.0005 | **0.7986** $\pm$ **0.0614** | 0.1754 $\pm$ 0.0026 | 0.1212 $\pm$ 0.0016 | 1.113 |
| $\text{MoAT}_{\text{w.o.CF}}$ | 0.0853 $\pm$ 0.0013 | 0.0290 $\pm$ 0.0005 | 0.0513 $\pm$ 0.0004 | 0.8565 $\pm$ 0.0840 | 0.1801 $\pm$ 0.0021 | 0.1263 $\pm$ 0.0023 | 5.938 |
| $\text{MoAT}_{\text{w.o.IP}}$ | 0.0855 $\pm$ 0.0018 | 0.0260 $\pm$ 0.0013 | 0.0539 $\pm$ 0.0016 | 0.8555 $\pm$ 0.0745 | 0.1864 $\pm$ 0.0053 | **0.1165** $\pm$ **0.0017** | 4.264 |
| $\text{MoAT}_{\text{w.o.OS}}$ | 0.0898 $\pm$ 0.0027 | 0.0258 $\pm$ 0.0008 | 0.0548 $\pm$ 0.0022 | 0.8196 $\pm$ 0.0709 | 0.1833 $\pm$ 0.0046 | 0.1180 $\pm$ 0.0030 | 4.387 |
| **MoAT** | 0.0816 $\pm$ 0.0016 | 0.0257 $\pm$ 0.0004 | 0.0494 $\pm$ 0.0006 | 0.8134 $\pm$ 0.0959 | **0.1727** $\pm$ **0.0020** | 0.1176 $\pm$ 0.0019 | - |

LightTS (Zhang et al., 2022) and DLinear (Zeng et al., 2023), as well as Transformer-based models including Autoformer (Wu et al., 2021), FEDformer (Zhou et al., 2022b), Pyraformer (Liu et al., 2021), Crossformer (Zhang & Yan, 2022), and PatchTST (Nie et al., 2022). Regarding multi-modal baselines, we include MM-Linear, MM-LSTM, and MM-TST. MM-Linear and MM-TST are designed based on extensions of DLinear and PatchTST, respectively, to integrate text information for time series forecasting. MM-LSTM is developed by modifying the multi-modal time series classification method (Deznabi et al., 2021) for forecasting purposes. For more details, refer to Appendix C.

**Experimental setup.** We partitioned each dataset into train/validation/test sets with a ratio of 6:2:2. Unless otherwise stated, MoAT and its baselines predicted the immediate subsequent step (i.e., $T = 1$) based on the preceding 8 timesteps (i.e., $L = 8$), while MoAT can forecast over longer durations (see Appendix D). By default, we used a hidden dimension of 64 and 4 attention heads for the Transformer encoder. For more specific configuration information, refer to Appendix B.4.

## 4.2 Q1. ACCURACY

We begin by evaluating the accuracy of MoAT in time series forecasting. As shown in Table 1, MoAT achieves a significant performance advantage over both uni-modal and multi-modal baselines. Specifically, MoAT achieves the lowest MSE across five out of six datasets, exhibiting improvements ranging from 6.455% to 71.688%, on average. This demonstrates the effectiveness of incorporating multi-modal data into the model, as well as the careful design choices incorporated into MoAT. For additional experimental results, refer to Appendix D.

## 4.3 Q2. EFFECTIVENESS

Next, we conduct ablation studies to validate the effectiveness of the design choices made in MoAT. The results are summarized in Table 2. For further details on each variant, refer to Appendix C.3.

**Effectiveness of multi-modality.** We study the effects of incorporating multi-modal data into time series forecasting. To this end, we examine two variants of MoAT, each utilizing one of the modalities: (1) $\text{MoAT}_{\text{time}}$ uses only the time series modality as input, and (2) $\text{MoAT}_{\text{text}}$ uses only the text modality as input. As shown in Table 2, MoAT outperforms both variants, enhancing the accuracy of $\text{MoAT}_{\text{time}}$ by 6.613% and $\text{MoAT}_{\text{text}}$ by 16.696%, on average. This demonstrates that under MoAT, both the time series and text modalities contribute to accurate time series forecasting.

**Effectiveness of dual augmentation.** MoAT consists of a multi-modal augmented encoder that utilizes dual augmentation methods on the input multi-modal data. To investigate the effectiveness of the augmentation schemes, we consider two variants: (1) $\text{MoAT}_{\text{sample}}$ augments the data in the sample space yielding single-modal representations $\hat{\mathbf{z}}_{\text{time}}$ and $\hat{\mathbf{z}}_{\text{text}}$, and (2) $\text{MoAT}_{\text{feature}}$ augments the data in the feature space generating cross-modal representations $\tilde{\mathbf{z}}_{\text{time}}$ and $\tilde{\mathbf{z}}_{\text{text}}$. As shown in Table 2, MoAT, which incorporates both augmentation schemes, forecasts time series more accurately than $\text{MoAT}_{\text{sample}}$ and $\text{MoAT}_{\text{feature}}$ by 0.724% and 1.532%, respectively, on average. This implies that both augmentation methods are effective, and adopting both provides the best performance.

**Effectiveness of other design choices.** We conduct ablation studies to verify other design components of MoAT including the discussions outlined in Section 3. To this end, we consider four variants: (1) $\text{MoAT}_{\text{w.o.JD}}$ (w.o. *joint decomposition*) forecasts without trend-seasonal decomposition, (2) $\text{MoAT}_{\text{w.o.CF}}$ (w.o. *cross-fusion*) removes the cross-fusion scheme from MoAT, only com-

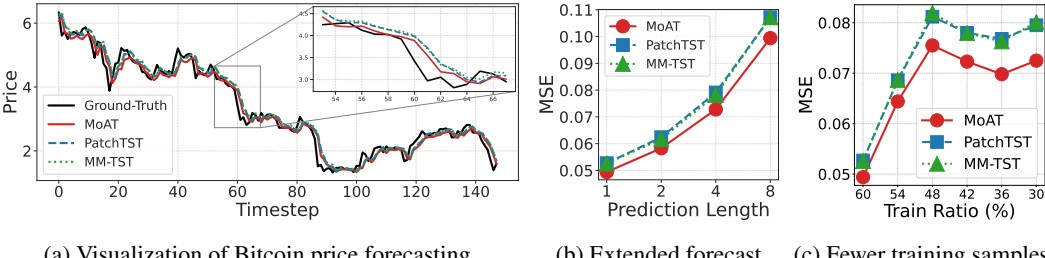

(a) Visualization of Bitcoin price forecasting     (b) Extended forecast    (c) Fewer training samples

Figure 5: (a) MoAT accurately predicts future Bitcoin price compared to the uni-modal PatchTST and multi-modal MM-TST time series forecasting models. (b) MoAT consistently outperforms its baselines for various forecasting lengths. (c) MoAT maintains superior performance over its baselines even when trained with a reduced number of training samples.

bining the corresponding representations from each component, (3) MoAT$_{\text{w.o.IP}}$ (w.o. *independent predictions*) averages the representations in each component and makes a united prediction, and (4) MoAT$_{\text{w.o.OS}}$ (w.o. *offline synthesis*) combines the prediction synthesis module into the main model, training its parameters in an end-to-end manner. More details about each variant can be found in Appendix C.3. As shown in Table 2, MoAT outperforms these variants, improving them by $1.113\%$, $5.938\%$, $4.264\%$, and $4.387\%$, respectively, confirming the effectiveness of each design choice.

We conduct a visual analysis of the distributions of representations obtained by MoAT using the Bitcoin dataset. Fig. 4 illustrates the distribution of eight distinct representations (Eq. 4) across the timesteps of the test set, in a 2D space using t-SNE (Van der Maaten & Hinton, 2008). It is evident that these eight independent representations cover a broader space compared to the united representation obtained by averaging. This observation provides insight into why MoAT outperforms its variant, MoAT$_{\text{w.o.IP}}$, which aggregates the representations for making a single prediction. Furthermore, it is

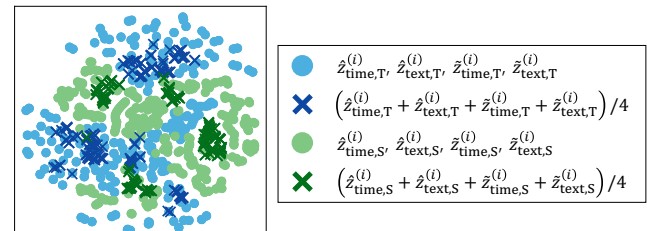

Figure 4: Augmentation space of MoAT. Multi-modal data augmentation expands the representation space, leading to more accurate forecasting of temporal dynamics.

noteworthy that the representations originating from the trend and seasonal components are clearly distinguishable, implying their complementary roles in making predictions.

## 4.4 Q3. VERSATILITY

We evaluate MoAT across different settings. In Fig. 5, we illustrate our results using the Bitcoin dataset as a case study. As shown in Fig. 5a, MoAT provides more accurate predictions of future Bitcoin prices compared to its strongest uni-modal and multi-modal competitors, PatchTST and MM-TST. In addition, as seen in Fig. 5b, MoAT consistently exhibits superior performance compared to its baselines when forecasting over longer future timesteps, specifically 2, 4, and 8 timesteps ahead. Moreover, Fig. 5c shows that MoAT maintains its superiority over its baselines even when trained with a reduced number of training samples, demonstrating the robustness and adaptability of MoAT across varying data availability scenarios. These results emphasize MoAT as a reliable choice for accurate time series forecasting in diverse settings.

## 5 CONCLUSIONS

This paper presents MoAT, a novel multi-modal augmented approach for time series forecasting aimed at overcoming data scarcity challenges. MoAT efficiently integrates information from various sources, including textual data, by combining both sample-wise and feature-wise multi-modal augmentation and fusion. Additionally, it enhances prediction accuracy through trend-seasonal decomposition using both data modalities. Extensive experiments conclusively demonstrate that MoAT surpasses existing methods, confirming its effectiveness and robustness in addressing data scarcity issues within the realm of time series forecasting.

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

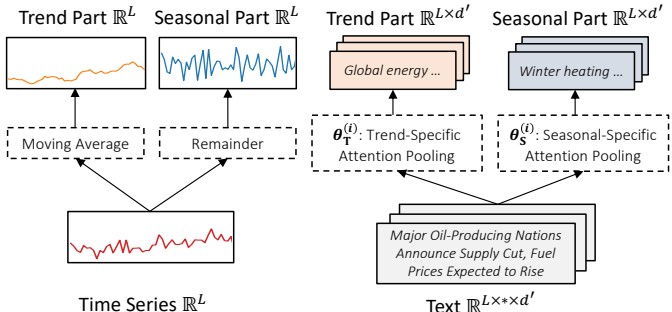

Figure 6: Trend-seasonal decomposition of time series and text data. For time series, MoAT applies the standard decomposition method, utilizing the moving average kernel and the residual component from the input time series. For text data, MoAT utilizes distinct attention parameters to aggregate text embeddings, allowing for varied emphasis on documents.

## A    FURTHER DETAILS OF MOAT

In this section, we provide additional details of the architecture of MoAT.

**Instance Normalization.** We employ the instance normalization scheme for the input time series, a choice inspired by its proven effectiveness in PatchTST (Nie et al., 2022). This involves computing the mean $m_i$ and standard deviation $\sigma_i$ for the input time series $\boldsymbol{x}^{(i)}$ of the $i$-th channel. Subsequently, we normalize the time series using these values, i.e., $(\boldsymbol{x}^{(i)} - m_i)/\sigma_i$. After obtainint the prediction $\hat{\boldsymbol{y}}^{(i)}$, we reverse this process through de-normalization using $\sigma_i \hat{\boldsymbol{y}}^{(i)} + m_i$. MoAT takes an additional step to account for the temporal locality within the time series. Specifically, instead of using the mean value $m_i$, it opts for $(m_i + \boldsymbol{x}_L^{(i)})/2$, which is the average between the mean value and the last value of the input time series $\boldsymbol{x}^{(i)}$. This adjustment enables the method to better capture the temporal locality during the forecasting process.

**Channel Independence.** Building upon previous studies such as CNN (Zheng et al., 2014), DLinear (Zeng et al., 2023), and PatchTST Nie et al. (2022), we incorporate the concept of channel-independence within the MoAT framework. In this approach, instead of mixing information across channels, we consider each channel as an independent data sample that shares the projection weight parameters and Transformer encoder weights. Thus, each input time series sample is considered a univariate time series along with its corresponding specialized text embeddings. We plan to explore channel-dependent version of MoAT in our future work.

**Trend-Seasonal Decomposition.** As described in Section 3, we adopt trend-seasonal decomposition of both time series and text in MoAT, as illustrated in Fig. 6.

For **time series**, we use the standard trend-seasonal decomposition method. This involves utilizing the moving average kernel on the input time series $\boldsymbol{x}^{(i)}$ to extract the trend component $\boldsymbol{x}_T^{(i)}$, while the residual becomes the seasonal component $\boldsymbol{x}_S^{(i)}$. Thus, $\boldsymbol{x}^{(i)} = \boldsymbol{x}_T^{(i)} + \boldsymbol{x}_S^{(i)}$ holds.

For **text data**, we introduce two independent sets of attention parameters $\theta_T^{(i)} = \{\mathbf{W}_T^{(i)}, \boldsymbol{b}_T^{(i)}, \mathbf{V}_T^{(i)}\}$ and $\theta_S^{(i)} = \{\mathbf{W}_S^{(i)}, \boldsymbol{b}_S^{(i)}, \mathbf{V}_S^{(i)}\}$ to extract trend-related and seasonal-related text information, respectively. More precisely, given a set of text embeddings $\boldsymbol{d}_{t_a:t_b} \in \mathbb{R}^{|\mathcal{D}_{t_a:t_b}| \times d'}$ ranging from $t_a$ to $t_b$, we apply attention pooling using $\theta_M^{(i)}$ to aggregate the embeddings as follows:

$$\mathbf{Softmax}\left(\mathbf{tanh}(\boldsymbol{d}_{t_a:t_b}\mathbf{W}_M^{(i)} + \boldsymbol{b}_M^{(i)})\mathbf{V}_M^{(i)}\right)\boldsymbol{d}_{t_a:t_b} \qquad (6)$$

for each component $\mathrm{M} \in \{\mathrm{T}, \mathrm{S}\}$. By using distinct sets of attention parameters, MoAT can selectively emphasize documents to extract information pertaining to trend and seasonal dynamics.

## B    Experimental Details

In this section, we share more detailed information about the experimental settings.

### B.1    Datasets

In this subsection, we detail our preprocessing methods for the multi-modal datasets employed in our experiments. To the best of our knowledge, currently, there is no benchmark time series dataset accompanied by textual information for each timestep. Thus, we collected and processed the raw datasets, and they will be released to the public.

**Fuel** is a monthly dataset consisting of gas [2] and oil [3] prices, spanning from January 2000 to September 2022. For each month, we extracted news articles from The New York Times [4] that contain relevant keywords {*brent crude, crude oil, energy policy, gas demand, gas market, gasoline price, natural gas, oil demand, oil market, oil price, OPEC*}, as suggested by ChatGPT.

**Metal** is a monthly dataset consisting of gold [5] and silver [6] prices, spanning from January 2000 to August 2022. For each month, we extracted news articles from The New York Times that contain relevant keywords {*coinage, COMEX, currency strength, exchange market, exchange rate, Fort Knox, gold and silver, gold coin, gold industry, gold market, gold mining, gold price, gold reserves, gold silver, gold standard, hedging, inflation, karat, LBMA, mining output, precious metal, quantitative easing, recession, safe-heaven asset, silver coins, silver industry, silver institute, silver market, silver price, sterling silver, supply chain disruptions, world gold council*}, as suggested by ChatGPT.

**Stock-Index** is a monthly dataset consisting of the commodity price index[7] ranges from March 2010 to Feburary 2022. The news articles related to each month are gathered from *S&P Global Commodity Insights*.

**Stock-Tech** is a weekly dataset of Microsoft (MSFT) and Apple (AAPL) stock prices ranging from December 1, 2006 to November 30, 2016. For each week, we aggregated news articles featuring these two companies retrieved from The New York Times API [8].

**Bitcoin** is a daily dataset consisting of Bitcoin (BTC), Ethereum (ETH), Tether (USDT), and Binance Coin (BNB) prices [9], spanning from November 13, 2017 to November 23, 2019. For each day, we filtered tweets about Bitcoins [10] that received at least 100 likes and 50 retweets.

**Covid** is a daily dataset consisting of the number of confirmed COVID-19 cases in 10 countries: China, the United States, Italy, Singapore, South Korea, Georgia, Japan, Canada, Russia, and Australia [11], spanning from January 22, 2020 to July 27, 2020. For each day, we used news content that is related to Coronavirus which was aggregated, analyzed, and enriched by AYLIEN using the AYLIEN's News Intelligence Platform. Among a large number of articles, we filtered 20 articles per day based on the number of times the articles were shared on social media platforms including Facebook, LinkedIn, Reddit, and Google Plus.

### B.2    Hyperparameter Search

We report the hyperparameter search space for each method, including MoAT. By default, a hidden dimension of 64 is used, and if needed, dropout with a probability of 0.2 is applied. We search for the optimal learning rate from {0.00005, 0.0001} and weight decay from {0.0001, 0.001}.

In the case of attention-based models (Autoformer, FEDformer, Pyraformer, Crossformer, PatchTST, MM-TST, and MoAT), we use 2 attention layers and explore the number of attention

---

[2] https://www.kaggle.com/datasets/psycon/historical-natural-gas-data-from-2000-to-202204

[3] https://www.kaggle.com/datasets/psycon/historical-brent-oil-price-from-2000-to-202204

[4] https://www.kaggle.com/datasets/aryansingh0909/nyt-articles-21m-2000-present

[5] https://www.kaggle.com/datasets/psycon/daily-gold-price-historical-data

[6] https://www.kaggle.com/datasets/psycon/daily-silver-price-historical-data

[7] https://www.indexmundi.com/commodities/

[8] https://www.kaggle.com/datasets/BidecInnovations/stock-price-and-news-realted-to-it

[9] https://www.kaggle.com/datasets/sudalairajkumar/cryptocurrencypricehistory

[10] https://www.kaggle.com/datasets/alaix14/bitcoin-tweets-20160101-to-20190329

[11] https://www.kaggle.com/datasets/imdevskp/corona-virus-report

Table 3: Summary of dataset statistics, including time series and text data. "Timesteps" refers to the time series length, "# of Channels" indicates the number of channels in the time series, and "Resolution" denotes the temporal granularity. "Min," "Max," and "Avg" represent the minimum, maximum, and average number of texts at each timestep, respectively.

| Dataset | Time Series | | | Text | | |
|---|---|---|---|---|---|---|
| | Timesteps | # of Channels | Resolution | Min. | Max | Avg. |
| Fuel | 273 | 2 | Month | 1 | 117 | 21.758 |
| Metal | 272 | 2 | Month | 3 | 187 | 34.033 |
| Stock-Index | 144 | 10 | Month | 12 | 518 | 289.611 |
| Stock-Tech | 522 | 2 | Week | 1 | 10 | 7.019 |
| Bitcoin | 741 | 4 | Day | 2 | 369 | 53.896 |
| Covid | 188 | 10 | Day | 20 | 20 | 20.000 |

heads from $\{4, 16\}$. For models utilizing time series decomposition (Autoformer, FEDformer, and MoAT), we search for the moving average for obtaining the trend component from $\{3, 5\}$. In patch-based methods (PatchTST, Crossformer, MM-TST, and MoAT), we used the patch length $P = 4$ and stride $S = 2$.

We determined the best configuration that resulted in the highest accuracy (i.e., lowest MSE) in the validation set. Using this identified hyperparameter setting, we report the average performance along with its standard deviation, calculated across performance from 10 random seeds.

### B.3 Implementation Details

**Base Implementation.** We developed MoAT and its variants based on the open source implementation for time series forecasting [12]. We used the Ridge regression provided by scikit-learn (Pedregosa et al., 2011) for the prediction synthesis module of MoAT.

**Channel Independence.** Following PatchTST, we employ a channel-independent approach for time series forecasting by considering each input time series as a univariate time series. Given a batch of $B$ samples sized $B \times C \times L$, where $C$ is the number of channels and $L$ is the number of past observable timesteps, we reshape the input as $(B \cdot C) \times L$ so that each channel within the batch is treated independently. After segmenting each univariate time series into $N$ patches, the input is reshaped to $(B \cdot C) \times N \times d$, which can be readily utilized by any standard Transformer architecture.

**Pretrained Language Model.** To generate embeddings for each text, we used the pretrained language model. Specifically, we used the sentence transformer provided by Hugging Face with the pretrained model named `all-mpnet-base-v2` [13], which is trained on a 1B sentence pairs dataset using contrastive loss.

### B.4 Training Details

**Data Split.** For each dataset, we split the time series and text data based on their temporal order into train, validation, and test sets using a 6:2:2 ratio. Following Zhou et al. (2021), the entire time series data is normalized using the mean and standard deviation of the training set.

**Training Epochs.** The total number of training epochs is set to 200. But, if the validation loss fails to decrease for 10 consecutive epochs, we stop the training process early. For the Steel dataset, we limit the training to just 1 epoch due to its small size.

**Offline Prediction Synthesis.** Our offline prediction synthesis module is trained separately from the main forecasting model. We utilize a Ridge regression approach, incorporating 16 weight parameters and an intercept for aggregating the 16 predictions from the decoder. During each training epoch, a Ridge regression is trained from scratch using the training set. Then, the model's learned parameters are utilized to evaluate validation and test sets.

---

[12] https://github.com/thuml/Time-Series-Library
[13] https://huggingface.co/sentence-transformers/all-mpnet-base-v2

## C  BASELINES AND VARIANTS

In this section, we describe the baseline methods that are compared with MoAT in Section 4. Each method is either uni-modal or multi-modal models.

### C.1  UNI-MODAL BASELINE METHODS

We first describe the uni-modal baselines we used to evaluate MoAT in Section 4.

**LightTS** (Zhang et al., 2022) is a simple MLP-based time series forecasting model. Specifically, it utilizes MLP-based structures to effectively capture two significant patterns, short-term and long-term temporal dependencies within the time series.

**DLinear** (Zeng et al., 2023) is a recent non-Transformer model that argues that the Transformer-based models have difficulty in capturing ordering information within the time series. It consists of two distinct linear layers which are independently applied to the trend and seasonal components of the input time series. The resulting outputs are then summed to generate the final prediction.

**Autoformer** (Wu et al., 2021) is a Transformer-based method that incorporates time series decomposition, inspired by classical time series analysis methods. In addition, it integrates an auto-correlation mechanism, which replaces the conventional self-attention layer within the Transformer.

**FEDformer** (Zhou et al., 2022b) is a Transformer-based method that also uses a time series decomposition scheme. Motivated by the fact that time series tend to have sparse representations, it is based on a frequency-enhanced Transformer, which offers linear complexity to the length of the input time series.

**Pyraformer** (Liu et al., 2021) is a Transformer-based method incorporating a pyramidal attention module. Specifically, the module consists of inter-scale connections that summarize features at various resolutions and intra-scale connections that capture temporal dependencies across diverse ranges. Moreover, it scales linearly to the length of the input time series.

**Crossformer** (Zhang & Yan, 2022) is a Transformer-based method that utilizes cross-dimension dependency for time series forecasting. Specifically, it segments the input time series into patches and trains a module that captures cross-time and cross-dimension dependencies among them.

**PatchTST** (Nie et al., 2022) is a Transformer-based method that adopts an independent learning approach for each channel using a shared encoder. This model demonstrates that Transformers are effective when the input time series is segmented into patches at the subseries level.

### C.2  MULTI-MODAL BASELINE METHODS

In this subsection, we describe MM-Linear, MM-LSTM, and MM-TST, baseline models designed to compare against MoAT. MM-Linear and MM-TST extend DLinear (Zeng et al., 2023) and PatchTST (Nie et al., 2022), respectively to integrate multi-modal data within their framework. MM-LSTM is an adapted version of a time series classification method (Deznabi et al., 2021), designed for mortality prediction within the healthcare domain.

Commonly, these models are given with a time series $\boldsymbol{x}^{(i)} = (x_1^{(i)}, \cdots, x_L^{(i)}) \in \mathbb{R}^L$ of the $i$-th channel and a set of texts $\mathcal{D}_t = \{\mathcal{D}_{t,1}, \cdots, \mathcal{D}_{t,|\mathcal{D}_t|}\}$ at each timestep $t = 1, \cdots, L$. Using the pretrained language model, each set of texts is represented as a set of text embeddings $\boldsymbol{d}_t = \{\boldsymbol{d}_{t,1}, \cdots, \boldsymbol{d}_{t,|\mathcal{D},t|}\} \in \mathbb{R}^{|\mathcal{D}_t| \times d'}$ where $\boldsymbol{d}_{t,j} = \text{PLM}(\mathcal{D}_{t,j}) \in \mathbb{R}^{d'}$.

**MM-Linear** is a multi-modal method that extends DLinear, a simple linear method for time series forecasting. The input time series $\boldsymbol{x}^{(i)}$ is decomposed into the trend part $\boldsymbol{x}_{\text{T}}^{(i)}$ and the seasonal part $\boldsymbol{x}_{\text{S}}^{(i)}$ from the seasonal-trend decomposition. For the text data, the method applies attention pooling (Eq. 6) to each set of texts at each timestep $t$ to derive a single aggregated text embedding $\boldsymbol{z}_{\text{text},t}^{(i)}$. Then, these text embeddings are averaged to obtain $\boldsymbol{z}_{\text{text}}^{(i)} = (\boldsymbol{z}_{\text{text},1}^{(i)} + \cdots + \boldsymbol{z}_{\text{text},L}^{(i)})/L \in \mathbb{R}^{d'}$, which represents the general textual information associated with the $i$-th channel. Finally, it concatenates time series and text embeddings, $\boldsymbol{x}_{\text{T}}^{(i)} \parallel \boldsymbol{z}_{\text{text}}^{(i)} \in \mathbb{R}^{L+d'}$ and $\boldsymbol{x}_{\text{S}}^{(i)} \parallel \boldsymbol{z}_{\text{text}}^{(i)} \in \mathbb{R}^{L+d'}$, which represent the multi-modal embeddings of the trend and seasonal dynamics, respectively. Following this, two

independent one-layer linear layers are individually applied to each component and summed to derive the final prediction, consistent with the original DLinear.

**MM-LSTM** is inspired by the time series classification model for patients' mortality prediction. It employs LSTM to capture the temporal dependencies within the time series $x$, resulting in $z_{\text{time}} = \text{LSTM}(x) \in \mathbb{R}^{d''}$. For text data, it averages all the text embeddings at timestep $t = 1, \cdots, L$ to obtain $z_{\text{text}} \in \mathbb{R}^{d'}$. Upon concatenating $z_{\text{time}}$ and $z_{\text{text}}$ to yield $z \in \mathbb{R}^{d'+d''}$, an independent one-layer linear layer is applied for each channel to generate a unique prediction for each channel. To ensure forecasting stability, the input time series is normalized with zero mean and unit standard deviation, and they are added back to the output prediction.

**MM-TST** is an extension of the Transformer-based time series forecasting model, PatchTST. In line with the original PatchTST, it produces patch-wise time series embeddings $z_{\text{time}}^{(i)} \in \mathbb{R}^{N \times d}$. In addition, it computes patch-wise text embeddings $z_{\text{text}}^{(i)} \in \mathbb{R}^{N \times d}$, employing the same approach as in MoAT. Next, temporally aligned patches from the two modalities are concatenated, forming a $2d$-dimensional patch at each timestep. Following this, a single linear-layer decoder is employed on the flattened output patches to predict $T$ future timesteps.

## C.3 VARIANTS OF MoAT

We provide the details of the variants of MoAT that are used to evaluate the effectiveness of the design components of MoAT in Section 4.3.

**Effectiveness of multi-modality.** To assess the effectiveness of integrating multi-modal datasets, we examine two variants of MoAT. The first, MoAT$_{\text{time}}$ exclusively utilizes the time series modality and incorporates trend-seasonal decomposition. On the other hand, MoAT$_{\text{text}}$ mainly relies on the text modality to forecast time series. Given the challenge of accurately predicting one modality using another, MoAT$_{\text{text}}$ leverages hints from time series data normalization described in Appendix A. That is, the generated prediction is de-normalized by the mean, the last value, and the standard deviation of the input time series. As these variants operate as single-modal methods, neither multi-modal augmentation nor cross-fusion can be applied.

**Effectiveness of dual augmentation.** To evaluate the effectiveness of the multi-modal data augmentation schemes, we explore two variants, MoAT$_{\text{sample}}$ and MoAT$_{\text{feature}}$. These variants integrate both time series and text data in their approach. First, MoAT$_{\text{sample}}$ employs sample-wise augmentation to generate $\hat{z}_{\text{time}}$ and $\hat{z}_{\text{text}}$. Second, MoAT$_{\text{feature}}$ utilizes feature-wise augmentation to generate $\tilde{z}_{\text{time}}$ and $\tilde{z}_{\text{text}}$. In each variant, two representations are generated for each trend and seasonal component, resulting in four predictions from cross-fusion. In contrast, MoAT generates 16 predictions by considering more representations for the trends and seasonal components by incorporating both augmentation schemes.

**Effectiveness of other design choices.** To validate the effectiveness of the key design choices discussed in Section 3, we examine three variants of MoAT: MoAT$_{\text{w.o.CF}}$, MoAT$_{\text{w.o.IP}}$, and MoAT$_{\text{w.o.OS}}$.

- **MoAT$_{\text{w.o.JD}}$** removes the trend-seasonal decomposition module from MoAT. Consequently, it generates four representations instead of eight, and the cross-fusion scheme is removed as well.
- **MoAT$_{\text{w.o.CF}}$** removes the cross-fusion scheme utilized in MoAT. That is, instead of considering all 16 combinations of aggregations of representations from $\mathbf{Z}_{\text{T}}^{(i)}$ and $\mathbf{Z}_{\text{S}}^{(i)}$, it only aggregates the corresponding representations. Specifically, it aggregates $\hat{z}_{\text{time,T}}^{(i)}$ & $\hat{z}_{\text{time,S}}^{(i)}$, $\hat{z}_{\text{text,T}}^{(i)}$ & $\hat{z}_{\text{text,S}}^{(i)}$, $\tilde{z}_{\text{time,T}}^{(i)}$ & $\tilde{z}_{\text{time,S}}^{(i)}$, and $\tilde{z}_{\text{text,T}}^{(i)}$ & $\tilde{z}_{\text{text,S}}^{(i)}$, resulting in four predictions.
- **MoAT$_{\text{w.o.IP}}$** generates a single prediction by aggregating the representations. More specifically, it computes a united representation for the trend and seasonal components, denoted as $\bar{z}_{\text{T}}^{(i)}$ and $\bar{z}_{\text{S}}^{(i)}$:

$$\bar{z}_{\text{T}}^{(i)} = \frac{\hat{z}_{\text{time,T}}^{(i)} + \hat{z}_{\text{text,T}}^{(i)} + \tilde{z}_{\text{time,T}}^{(i)} + \tilde{z}_{\text{text,T}}^{(i)}}{4} \quad \text{and} \quad \bar{z}_{\text{S}}^{(i)} = \frac{\hat{z}_{\text{time,S}}^{(i)} + \hat{z}_{\text{text,S}}^{(i)} + \tilde{z}_{\text{time,S}}^{(i)} + \tilde{z}_{\text{text,S}}^{(i)}}{4}.$$

Using these unified representations of both trend and seasonal components, it generates a singular, united prediction by summing them:

$$\mathcal{L}^{(i)} = \left\| \bar{z}_{\text{T}}^{(i)} \mathbf{W}_{\text{dec}}^{\text{T}} + \bar{z}_{\text{S}}^{(i)} \mathbf{W}_{\text{dec}}^{\text{S}} - \mathbf{Y}^{(i)} \right\|_2^2$$

Table 4: Time series forecasting accuracy (in terms of MSE) in six real-world datasets for a prediction length of $T = 2$. The best results are in **bold** and the second best are underlined. The average improvements (%) and ranks are reported on the right. MoAT performs the best on average.

| Method | Fuel | Metal | Bitcoin | Stock-Index | Covid | Stock-Tech | Improv. | Rank |
|---|---|---|---|---|---|---|---|---|
| LightTS | $0.1707 \pm 0.0318$ | $0.1609 \pm 0.1045$ | $0.1853 \pm 0.0943$ | $1.6850 \pm 0.3808$ | $0.4022 \pm 0.1285$ | $0.7412 \pm 0.3926$ | 58.72 | 8.66 |
| DLinear | $0.2335 \pm 0.0390$ | $0.0870 \pm 0.0262$ | $0.0911 \pm 0.0199$ | $\mathbf{0.8116} \pm \mathbf{0.0500}$ | $0.2783 \pm 0.0950$ | $0.4051 \pm 0.0919$ | 37.54 | 7.16 |
| Autoformer | $0.1948 \pm 0.0220$ | $0.0669 \pm 0.0066$ | $0.0957 \pm 0.0055$ | $0.8549 \pm 0.0181$ | $0.2672 \pm 0.0175$ | $0.2905 \pm 0.242$ | 31.82 | 6.50 |
| FEDformer | $0.1464 \pm 0.0055$ | $0.0579 \pm 0.0027$ | $0.0897 \pm 0.0091$ | $0.8487 \pm 0.0181$ | $0.2438 \pm 0.0126$ | $0.1808 \pm 0.0059$ | 19.10 | 4.83 |
| Pyraformer | $0.1783 \pm 0.0229$ | $0.2918 \pm 0.1485$ | $1.4690 \pm 0.2101$ | $1.3141 \pm 0.5165$ | $1.6353 \pm 0.2116$ | $11.8653 \pm 1.4095$ | 73.03 | 9.83 |
| Crossformer | $\underline{0.1330} \pm 0.0042$ | $0.1686 \pm 0.1029$ | $0.5108 \pm 0.0511$ | $1.6567 \pm 0.3210$ | $0.5584 \pm 0.0371$ | $8.2583 \pm 0.8155$ | 64.50 | 8.50 |
| PatchTST | $0.1375 \pm 0.0056$ | $0.0410 \pm 0.0013$ | $0.0624 \pm 0.0010$ | $0.8666 \pm 0.0178$ | $\mathbf{0.1966} \pm \mathbf{0.0087}$ | $\underline{0.1723} \pm 0.0041$ | 5.33 | $\underline{3.16}$ |
| MM-Linear | $0.1915 \pm 0.0126$ | $0.0547 \pm 0.0096$ | $\underline{0.0588} \pm 0.0023$ | $1.7323 \pm 0.0609$ | $0.2121 \pm 0.0194$ | $0.3559 \pm 0.0774$ | 29.94 | 6.00 |
| MM-LSTM | $0.2174 \pm 0.0128$ | $0.0686 \pm 0.0004$ | $0.0621 \pm 0.0023$ | $0.8720 \pm 0.0139$ | $0.2170 \pm 0.0013$ | $0.3201 \pm 0.0494$ | 25.95 | 6.16 |
| MM-TST | $0.1369 \pm 0.0028$ | $\underline{0.0402} \pm 0.0008$ | $0.0628 \pm 0.0016$ | $0.9076 \pm 0.0573$ | $0.2050 \pm 0.0156$ | $0.1731 \pm 0.0049$ | 6.59 | 3.83 |
| **MoAT** | $\mathbf{0.1215} \pm \mathbf{0.0024}$ | $\mathbf{0.0370} \pm \mathbf{0.0004}$ | $\mathbf{0.0583} \pm \mathbf{0.0005}$ | $\underline{0.8319} \pm 0.0496$ | $\underline{0.2011} \pm 0.0032$ | $\mathbf{0.1682} \pm \mathbf{0.0020}$ | - | **1.33** |

It is important to note that this approach differs from MoAT's loss function in Eq. 5, where each representation is used independently to derive the loss, and the losses are averaged instead of averaging the representations themselves.

- **MoAT$_{\text{w.o.OS}}$** optimizes the weight parameters $\lambda_1, \cdots, \lambda_{16}$ in an end-to-end fashion, concurrently with the training of the encoder and decoder. In this integrated training approach, the same optimizer is employed, enabling the model to simultaneously serve both the time series forecasting and prediction aggregating functionalities. Specifically, given the synthesized prediction $\hat{\mathbf{Y}}^{(i)}$ using $\lambda_1, \cdots, \lambda_{16}$, the final loss is the sum of Eq. 5 and the MSE loss of the synthesized one:

$$\mathcal{L}^{(i)} = \sum_{\mathbf{z}_{\text{T}}^{(i)} \in \mathbf{Z}_{\text{T}}^{(i)}} \sum_{\mathbf{z}_{\text{S}}^{(i)} \in \mathbf{Z}_{\text{S}}^{(i)}} \left\| \mathbf{z}_{\text{T}}^{(i)} \mathbf{W}_{\text{dec}}^{\text{T}} + \mathbf{z}_{\text{S}}^{(i)} \mathbf{W}_{\text{dec}}^{\text{S}} - \mathbf{Y}^{(i)} \right\|_2^2 + \left\| \hat{\mathbf{Y}}^{(i)} - \mathbf{Y}^{(i)} \right\|_2^2$$

## D  ADDITIONAL EXPERIMENTAL RESULTS

In this section, we present supplementary experimental results that are not included in the main content due to space constraints.

**Varying prediction lengths.** In Section 4, we utilized a default prediction length of $1$. In this section, we present the experimental results for varying prediction lengths, specifically at lengths of $2$, $4$, and $8$. The forecasting accuracy of MoAT and its baselines when predicting future $2$, $4$, and $8$ timesteps is depicted in Tables 4, 5, and 6, respectively. These results demonstrate that MoAT accurately predicts both short-term and longer-term future time series.

**Temporal alignment.** In Figure 7, we observe temporal similarities among representations across different timesteps in the Bitcoin dataset. It is noticeable that the similarity heatmaps for trend and seasonal representations exhibit comparable patterns, implying the temporal alignment within each trend and seasonal component. In addition, we can clearly see that heatmaps are distinguished between the two components, suggesting that each component offers unique temporal information that is potentially beneficial for accurate time series forecasting.

Table 5: Time series forecasting accuracy (in terms of MSE) in six real-world datasets for a prediction length of $T = 4$. The best results are in **bold** and the second best are underlined. The average improvements (%) and ranks are reported on the right. MoAT performs the best on average.

| Method | Fuel | Metal | Bitcoin | Stock-Index | Covid | Stock-Tech | Improv. | Rank |
|---|---|---|---|---|---|---|---|---|
| LightTS | 0.1865 ± 0.0101 | 0.1833 ± 0.1259 | 0.1519 ± 0.0503 | 1.8960 ± 0.3034 | 0.4079 ± 0.0737 | 1.3437 ± 1.0994 | 49.80 | 8.16 |
| DLinear | 0.3001 ± 0.0310 | 0.1307 ± 0.0222 | 0.1326 ± 0.0337 | 0.9719 ± 0.0603 | 0.3465 ± 0.0499 | 0.5073 ± 0.0469 | 36.95 | 7.66 |
| Autoformer | 0.2459 ± 0.0144 | 0.0853 ± 0.0062 | 0.1132 ± 0.0027 | 0.9969 ± 0.0270 | 0.2986 ± 0.0097 | 0.4003 ± 0.0213 | 25.36 | 6.16 |
| FEDformer | 0.1990 ± 0.0084 | 0.0814 ± 0.0062 | 0.1085 ± 0.0068 | 1.0194 ± 0.0329 | 0.2814 ± 0.0151 | 0.2805 ± 0.0261 | 16.16 | 5.00 |
| Pyraformer | 0.2247 ± 0.0259 | 0.4750 ± 0.2248 | 1.6298 ± 0.1754 | 1.7009 ± 0.1172 | 1.7994 ± 0.2905 | 13.1298 ± 1.5395 | 72.98 | 10.00 |
| Crossformer | 0.1998 ± 0.0059 | 0.3736 ± 0.1157 | 0.6778 ± 0.0672 | 1.6747 ± 0.1742 | 0.6559 ± 0.0431 | 9.2539 ± 0.6315 | 65.48 | 9.00 |
| PatchTST | 0.1981 ± 0.0027 | 0.0616 ± 0.0010 | 0.0791 ± 0.0010 | 0.9279 ± 0.0236 | **0.2427** ± 0.0126 | **0.2660** ± 0.0017 | 3.33 | 2.50 |
| MM-Linear | 0.2384 ± 0.0102 | 0.0888 ± 0.0258 | 0.0743 ± 0.0022 | 1.8208 ± 0.0548 | 0.3018 ± 0.0273 | 0.5122 ± 0.0694 | 29.52 | 7.00 |
| MM-LSTM | 0.2576 ± 0.0022 | 0.0855 ± 0.0006 | 0.0836 ± 0.0064 | 0.9416 ± 0.0034 | 0.2625 ± 0.0018 | 0.4312 ± 0.0015 | 20.11 | 5.50 |
| MM-TST | 0.1973 ± 0.0040 | 0.0596 ± 0.0008 | 0.0788 ± 0.0012 | 1.0192 ± 0.0181 | 0.2709 ± 0.0169 | 0.2731 ± 0.0054 | 6.41 | 3.33 |
| **MoAT** | **0.1687** ± 0.0014 | **0.0582** ± 0.0008 | **0.0728** ± 0.0007 | **0.9228** ± 0.0343 | 0.2508 ± 0.0065 | 0.2807 ± 0.0086 | - | **1.66** |

Table 6: Time series forecasting accuracy (in terms of MSE) in six real-world datasets for a prediction length of $T = 8$. The best results are in **bold** and the second best are underlined. The average improvements (%) and ranks are reported on the right. MoAT performs the best on average.

| Method | Fuel | Metal | Bitcoin | Stock-Index | Covid | Stock-Tech | Improv. | Rank |
|---|---|---|---|---|---|---|---|---|
| LightTS | 0.3006 ± 0.0118 | 0.2504 ± 0.0500 | 0.2573 ± 0.0528 | 2.0410 ± 0.1924 | 0.5448 ± 0.0416 | 2.1131 ± 0.6886 | 50.09 | 8.16 |
| DLinear | 0.3966 ± 0.0301 | 0.2160 ± 0.0430 | 0.2041 ± 0.0125 | 0.9518 ± 0.0393 | 0.5294 ± 0.0644 | 0.6941 ± 0.0383 | 34.04 | 8.00 |
| Autoformer | 0.3461 ± 0.0092 | 0.1265 ± 0.0064 | 0.1398 ± 0.0051 | 1.0202 ± 0.0184 | 0.3886 ± 0.0139 | 0.5799 ± 0.0101 | 18.22 | 6.33 |
| FEDformer | 0.3316 ± 0.0089 | 0.1100 ± 0.0000 | 0.1438 ± 0.0075 | 1.0138 ± 0.0396 | 0.3724 ± 0.0235 | 0.4907 ± 0.0533 | 12.70 | 4.66 |
| Pyraformer | 0.3068 ± 0.0201 | 0.6715 ± 0.0863 | 1.7596 ± 0.1932 | 1.3721 ± 0.4393 | 1.9689 ± 0.2832 | 14.6437 ± 1.4263 | 67.71 | 9.16 |
| Crossformer | 0.3318 ± 0.0175 | 0.4295 ± 0.1315 | 0.7878 ± 0.0876 | 1.4830 ± 0.2267 | 0.6923 ± 0.0796 | 10.6312 ± 0.8214 | 61.33 | 9.16 |
| PatchTST | 0.3358 ± 0.0050 | 0.0928 ± 0.0010 | 0.1072 ± 0.0006 | 0.8777 ± 0.0270 | 0.3645 ± 0.0131 | **0.4630** ± 0.0115 | 2.84 | 3.33 |
| MM-Linear | 0.3334 ± 0.0236 | 0.1441 ± 0.0950 | 0.1850 ± 0.0550 | 1.8349 ± 0.0064 | 0.4321 ± 0.0390 | 0.6679 ± 0.0403 | 31.96 | 7.50 |
| MM-LSTM | 0.3349 ± 0.0012 | 0.1287 ± 0.0006 | 0.1157 ± 0.0082 | 0.9423 ± 0.0372 | 0.3696 ± 0.0009 | 0.5770 ± 0.0150 | 13.44 | 5.00 |
| MM-TST | 0.3273 ± 0.0033 | 0.0919 ± 0.0011 | 0.1069 ± 0.0011 | 0.9359 ± 0.0462 | 0.4107 ± 0.0167 | 0.4673 ± 0.0053 | 5.37 | 3.16 |
| MoAT | **0.2740** ± 0.0029 | **0.0867** ± 0.0003 | **0.0994** ± 0.0005 | **0.8741** ± 0.0177 | **0.3644** ± 0.0087 | 0.5355 ± 0.0088 | - | **1.50** |

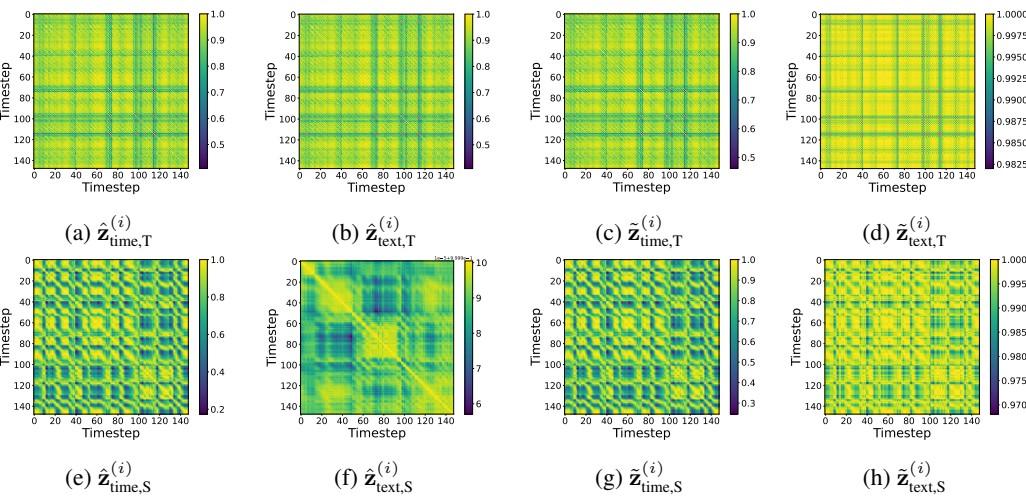

(a) $\hat{\mathbf{z}}_{\text{time,T}}^{(i)}$    (b) $\hat{\mathbf{z}}_{\text{text,T}}^{(i)}$    (c) $\tilde{\mathbf{z}}_{\text{time,T}}^{(i)}$    (d) $\tilde{\mathbf{z}}_{\text{text,T}}^{(i)}$

(e) $\hat{\mathbf{z}}_{\text{time,S}}^{(i)}$    (f) $\hat{\mathbf{z}}_{\text{text,S}}^{(i)}$    (g) $\tilde{\mathbf{z}}_{\text{time,S}}^{(i)}$    (h) $\tilde{\mathbf{z}}_{\text{text,S}}^{(i)}$

Figure 7: Pairwise cosine similarities between different timesteps for each of the eight representations. The representations of the trend component ($\mathbf{Z}_{\text{T}}^{(i)} = \{\hat{\mathbf{z}}_{\text{time,T}}^{(i)}, \hat{\mathbf{z}}_{\text{text,T}}^{(i)}, \tilde{\mathbf{z}}_{\text{time,T}}^{(i)}, \tilde{\mathbf{z}}_{\text{text,T}}^{(i)}\}$) and those of the seasonal component ($\mathbf{Z}_{\text{S}}^{(i)} = \{\hat{\mathbf{z}}_{\text{time,S}}^{(i)}, \hat{\mathbf{z}}_{\text{text,S}}^{(i)}, \tilde{\mathbf{z}}_{\text{time,S}}^{(i)}, \tilde{\mathbf{z}}_{\text{text,S}}^{(i)}\}$) exhibit distinctive temporal similarities, effectively capturing specific aspects essential for accurate time series forecasting.

