# OpenReview forum: "MoAT: Multi-Modal Augmented Time Series Forecasting"
_ICLR.cc/2024/Conference — Submitted to ICLR 2024_

### Official Review · Reviewer_roN3 · 2023-10-19

**Soundness:** 2 fair
**Presentation:** 3 good
**Contribution:** 3 good
**Rating:** 5
**Confidence:** 4

**Summary:**

This paper introduces MoAT (Multi-Modal Augmented Time Series Forecasting), an approach that leverages multimodal data, particularly text, to enhance time series forecasting by addressing data scarcity. In MoAT, text information is embedded into hidden vectors using a pretrained language model and aggregated into patches similar to time series patches. These patched time series and text data are then fed into a multi-modal augmented encoder that combines sample-wise and feature-wise augmentation methods to enrich multimodal representation learning. A joint trend-seasonal decomposition process is employed to capture underlying patterns in the data. The paper pairs all four representations (feature or sample, trend or season) of the two modalities (time series and text) into 16 combinations to make the final prediction. Extensive experiments conducted on real-world datasets demonstrate MoAT's effectiveness compared to previous state-of-the-art methods for single-modal time series forecasting.

**Strengths:**

- The concept of using multimodal data to tackle data scarcity and enhance time series forecasting is innovative and holds significant promise.
- The datasets collected and soon to be released in this paper will contribute positively to the community.
- The multi-modal augmented encoder, combining sample-wise and feature-wise augmentation, is an interesting and reasonable approach.

**Weaknesses:**

- It is not appropriate to directly transfer methods used for processing time series to text data:
   1. As a single value at a specific timestamp provides little information, PatchTST and Crossformer patch time series to form informative tokens. Text data already contains a wealth of information, not to mention that there are multiple texts at each time step, so using patching is unreasonable here.
   2. The decomposition of text data is unclear, particularly the definitions of "trend" and "season" for text. Equation (6) shows that the so-called trend-seasonal decomposition is just the same attention pooling with different sets of parameters, raising doubts about its ability to capture trend and seasonal dynamics as claimed.
- Datasets and baselines used in experiments are not so propoer:
  1. Table 3 shows that the largest dataset, Bitcoin， contains only 741 * 4 = 2,964 scalars， while the smallest contains less than 1,000. This raises concerns about the suitability of training complex neural networks with such limited data.
  2. The main experiment focuses on an input8-output1 setting, with both input and output series being very short. While most selected baselines are for longer term forecasting, i.e. they perform at least input96-output96 task. So the comparison is not so fair.

**Questions:**

1. It is advisable to conduct experiments to validate the necessity of using patching for text data.
2. Could you clarify the meaning of "trend" and "season" in the context of text data? Additionally, please elaborate on how the pooling in Equation (6), without further constraints, extracts trend and season.
3. Given the small dataset size and short input/output lengths, it is recommend to add traditional and straightforward models as baselines, such as: 1)repeat last: just repeating the last timestamp's value $x_{L}$ as prediction; 2)Vector autoregressive moving average; 3)DeepAR.

---

> ### Author Response · Authors · 2023-11-19
> **Dear Reviewer roN3: Thank you for the review**
>
> We appreciate your thoughtful comments, and we have made an effort to address them below.

---

> > ### Author Response · Authors · 2023-11-19
> > **Response to the weakness and questions**
> >
> > **C1. Text already contains a wealth of information, so using patching is unreasonable here.**
> >
> > As the reviewer commented, text provides sufficient information at each timestep while time-series data does not. However, we would like to emphasize the necessity of patching texts in our architecture. Specifically, we aggregate text embeddings within corresponding time intervals as the time series patches. This alignment is particularly crucial for feature-wise augmentation, where we concatenate temporally aligned time series and text patches as input for the shared multi-modal augmented encoder of MoAT.
> >
> > **C2. Decomposition of text data is unclear.**
> >
> > We acknowledge the reviewer's comment and modified the attention pooling function for texts by using time series patches as query vectors during the attention computation for text aggregation. From the modification, texts are aggregated differently based on the seasonal and trend time sereis patches. For more details, please refer to the common response above.
> >
> > **C3. Table 3 shows that the largest dataset, Bitcoin, contains only 3K scalars. This raises concerns about the suitability of training complex neural networks with such limited data.**
> >
> > As noted by the reviewer, the datasets we collected comprise a limited number of time series data points. To mitigate this scarcity in time series data, we employ multi-modal information, particularly leveraging the abundance of information within the text data. Specifically, text data contains a substantial amount of information that supplements the limited time series data. Moreover, the superiority of transformer-based models (e.g., PatchTST and Crossformer) over simpler linear models (e.g., DLinear and LightTS) suggests that complex neural networks like transformers indeed outperform simpler models within our datasets.
> >
> > **C4. Both input and output series is very short. While most selected baselines are for longer term forecasting. So the comparison is not so fair. It is recommended to add traditional and straightforward models as baselines.**
> >
> > While we incorporated basic linear models like LightTS and DLinear as baselines, we appreciate the reviewer’s feedback regarding the need for a fair comparison. We will take this into account and include baseline models to ensure a more comprehensive evaluation.

---

### Official Review · Reviewer_kpLg · 2023-10-26

**Soundness:** 2 fair
**Presentation:** 3 good
**Contribution:** 3 good
**Rating:** 5
**Confidence:** 4

**Summary:**

This paper introduces an interesting method for enhancing time-series forecasting by incorporating textual data. It applies both feature-wise and sample-wise augmentation techniques and integrates information about trend and seasonality to improve prediction accuracy. The authors have also contributed to the research community by publishing a new multimodal time-series and textual dataset.

**Strengths:**

1. The study introduces a new multimodal forecasting framework that integrates sample/feature-wise augmentation, cross-modal fusion, and seasonal-trend decomposition.
2. A new multimodal time-series and text dataset is presented as a contribution to the field.
3. The efficacy of the proposed approach is validated through experiments on six multimodal datasets.

**Weaknesses:**

1. It requires further clarification why the proposed textual decomposition components map to the trend and seasonality aspects of the time-series data.
2. The paper focuses on short-term forecasting, with the horizons in the experiments and even in the Appendix being relatively short when compared to existing benchmarks that typically extend from 96 to 712 timesteps.
3. The datasets primarily feature monthly or weekly sampling intervals. This raises concerns about the applicability of the approach to datasets with higher frequency sampling, such as hourly, where acquiring corresponding textual information could be challenging.
4. In Table 2, the MoAT_time model, even without textual data, appears to outperform many baseline models. The specific factors contributing to this enhanced performance are not adequately explained.
5. The results in Table 2 suggest that dual augmentation does not significantly enhance performance, questioning its effectiveness in the proposed framework.

**Questions:**

See Weaknesses above.

---

> ### Author Response · Authors · 2023-11-19
> **Dear Reviewer kpLg: Thank you for the review**
>
> We appreciate your thoughtful comments. Here, we have attempted to respond to your concerns as outlined below.

---

> > ### Author Response · Authors · 2023-11-19
> > **Response to the weakness and questions**
> >
> > **C1. It requires further clarification why the proposed textual decomposition components map to the trend and seasonality aspects of the time-series data.**
> >
> > We acknowledge the reviewer's comment and modified the attention pooling function for texts by using time series patches as query vectors during the attention computation for text aggregation. From the modification, texts are aggregated differently based on the seasonal and trend time sereis patches. For more details, please refer to the common response above.
> >
> > **C2. The paper focuses on short-term forecasting.**
> >
> > As commented by the reviewer, we intentionally chose a shorter forecasting length compared to some other time series forecasting works. This decision stems from the fact that the considered datasets are relatively smaller than benchmark datasets, and opting for a longer lookback or forecasting window inevitably leads to a reduction in the number of available training samples, potentially resulting in a significant performance decline. In addition, we would like to emphasize that forecasting time series even with a shorter lookback window presents a new challenge compared to existing works, which is often more practical and thus important.
> >
> > **C3. The datasets primarily feature monthly or weekly sampling intervals. This raises concerns about the applicability of the approach to datasets with higher frequency sampling.**
> >
> > As raised by the reviewer, testing on datasets with higher frequency is indeed an important consideration. However, we would like to emphasize that we collected six distinct datasets encompassing various sampling intervals, ranging from daily to monthly frequencies. In addition, elevating the sampling frequency can potentially introduce timesteps with missing text data, posing an additional challenge that requires careful attention. In our study, we assumed that at least one text is available at each timestep and leaved addressing this challenge as our future work.
> >
> > **C4. In Table 2, MoAT_time, even without textual data, appears to outperform many baseline models.**
> >
> > Thank you for highlighting this valuable insight. As noted by the reviewer, MoAT_time demonstrates superior performance compared to most baseline methods. We conjecture that his advantage can be attributed to the combined utilization of decomposition and patching within our framework. While there are existing time series forecasting methods that solely employ decomposition (e.g., DLinear and Autoformer) or patching (e.g., PatchTST and Crossformer), to our knowledge, none of these methods have incorporated both components simultaneously. In contrast, MoAT_time integrates and leverages both decomposition and patching to maximize their effectiveness.
> >
> > **C5. The results in Table 2 suggest that dual augmentation does not significantly enhance performance.**
> >
> > We would like to clarify that each variant of MoAT in Table 2 is fine-tuned within the same hyperparameter search space. The comparison between MoAT_sample and MoAT_feature aims to examine which augmentation method contributes more significantly. MoAT represents a comprehensive framework that encompasses the overall approach.

---

### Official Review · Reviewer_Moxw · 2023-10-31

**Soundness:** 2 fair
**Presentation:** 3 good
**Contribution:** 2 fair
**Rating:** 5
**Confidence:** 3

**Summary:**

The authors propose MoAT, multi-modal augmented general time series forecasting model that leverages multi-modal cross fusion and prediction synthesis

**Strengths:**

- originality: new application of multimodal time series for general time series forecasting
- quality: extensive experiments with good results, many ablations
- clarity: well-written paper, good structure
- significance: time series forecasting and foundation models are timely and relevant

**Weaknesses:**

- Lack of experiments assessing whether the model performs well with scarce data, which is painted as the main motivation of MoAT. Furthermore, figure 5c does not seem to corroborate the story that MoAT performs significantly better than other methods with data scarcity (hard to say without variance). MoAT still seems to derive its main performance improvements from increasing the train ratio.

- Lack of details in the caption of the T-SNE decomposition between time series and texts.

- The remarks about information uniqueness of cross-modal vs unimodal representations are not backed up, no reason for their contained information to be unique.

- Not obvious that the text data trend-seasonal decomposition actually decomposes into trend and seasonal data, it seems like you just use two sets of attention parameters. How do you actually get these to attend to either trend or seasonal information in the texts? This just seems like it introduces more parameters into the model.

- In fact, there is no comparison of model sizes and various scaling parameters for different methods. If you don't normalize, how do you know your performance increases aren't simply due to scaling up model size?

- Unclear empirical design for hyperparameter tuning. Why default at hidden dim of 64? What does if mean dropout =0.2 "if needed"? Why is the search for optimal learning rates and decay across two values each? If you're limited by compute or have a lot of hyperparameters, random search could be better than grid search.

- Formatting needs more consistency (e.g. "Fig." vs "Figure", figure 5 before figure 4, etc.)

**Questions:**

- how is text data decomposed into trend and seasonal components?

- What does "(non-)overlapping patches" mean? Clearly patches are overlapping if they share S values.

- Why are the texts unordered at each timestep? Is this a feature of the dataset used, or a design choice to ignore some of the granularity of the timestep?

- Why channel-specific parameters? Channel independence is a strong assumption.

- Is the forecasting in Figure 5 a autoregressive? What are you providing as inputs at each time step? What are the document inputs when autoregressively predicting? The visualization in figure 5 a is so zoomed out as to be uninformative (hard to tell the difference between the methods).

- the motivation for offline prediction synthesis discusses improved parameter efficiency, referencing Liang et al., 2022, but parameter efficiency is not discussed in the rest of the paper, nor when comparing methods? Furthermore, why is modularity desirable in this setting?

- Any intuition as to why MoAT_{time} is the best performer on the Fuel dataset?

- The ablations in table 2 suggest that the augmentations are not really helpful, considering the relative improvement compared to MoAT_{time} and MoAT_{text}. MoAT_{sample} seems pretty performant itself, being the best on Metal and Bitcoin, and second best on Fuel and Covid.

- Ridge regression includes a loss penalty. What weight did you choose for this?

- Why are the prediction lengths so short?


Overall, I'm giving this a 5 before discussion, as the storyline does not seem to align with the experiments. I am happy to amend my score following discussions.

---

> ### Author Response · Authors · 2023-11-19
> **Dear Reviewer Moxw: Thank you for the reviews**
>
> We are grateful for the time and effort you have invested in offering constructive feedback.

---

> ### Author Response · Authors · 2023-11-19
> **Responses to the weakness and questions #1**
>
> **C1. Lack of experiments assessing whether the model performs well with scarce data.**
>
> We would like to highlight that it is common in real-world scenarios to encounter time series datasets with a limited number of data samples or lengths, like the datasets collected for this study. Thus, our experimental results demonstrate the effectiveness of MoAT, in scenarios where datasets exhibit scarce time series information.
>
> **C2. Lack of details in the caption of the T-SNE decomposition between time series and texts.**
>
> We apologize for the lack of clarity in the caption for Figure 4. The colors blue and green in the figure represent the trend and seasonal representations, respectively. In the figure, “O”s depict the four representations acquired through the encoder, while “X”s indicate the averaged value of these four representations. A visual observation reveals that utilizing the four representations as individual data samples, instead of merging them, expands the representation space, which potentially contributes to more accurate and robust time series forecasting.
>
> **C3. The remarks about the information uniqueness of cross-modal vs. uni-modal representations are not backed up, no reason for their contained information to be unique.**
>
> We intended to convey that cross-modal and uni-modal representations are distinct from each other and may encompass different semantics, contributing to providing complementary information. An example is shown in Figure 1 (c) where time series and text present distinct information. We will refine this statement in the paper to avoid misunderstanding.
>
> **C4. Not obvious that the text data trend-seasonal decomposition actually decomposes into trend and seasonal data.**
>
> We acknowledge the reviewer's comment and modified the attention pooling function for texts by using time series patches as query vectors during the attention computation for text aggregation. From the modification, texts are aggregated differently based on the seasonal and trend time sereis patches. For more details, please refer to the common response above.
>
> **C5. There is no comparison of model sizes and various scaling parameters for different methods.**
>
> We conducted experiments on MoAT using various dimensions and observed that it is not significantly affected by the number of parameters of the model. We will include more detailed results to support this finding.
>
> **C6. What does (non-)overlapping patches mean? Clearly patches are overlapping if they share S values.**
>
> As the reviewer commented, patches are considered non-overlapping when S equals zero. We have phrased it as “(non-)overlapping” to highlight the flexibility in selecting S, which can be any non-negative integer smaller than the patch length.
>
> **C7. Why are the texts unordered at each timestep?**
>
> At every timestep, there could exist multiple texts, which can indeed be temporally ordered, as noted by the reviewer. In our current approach, we focused on the temporal granularity of the time series and did not account for the specific ordering of texts within each timestep. Nevertheless, we find the reviewer’s suggestion to be an insightful perspective and acknowledge its potential direction for future work.
>
> **C8. Why channel-specific parameters? Channel independence is a strong assumption.**
>
> We adopted channel-specific parameters for attention pooling in text aggregation due to the varying emphasis different channels may place on distinct texts. For example, when predicting the stock prices of various companies, each representing a different channel in time series data, the relevance and importance of texts could differ significantly. Thus, channel-specific parameters enable tailored aggregation methods, ensuring that diverse texts are aggregated differently based on their relevance to each specific channel’s prediction task. Regarding channel independence, a recent ICLR 2023 paper, PatchTST, demonstrated the powerfulness of channel-independence over channel-dependence. We followed their claims in our framework.

---

> > ### Author Response · Authors · 2023-11-19
> > **Responses to the weakness and questions #2**
> >
> > **C9. Is the forecasting in Figure 5 autoregressive?**
> >
> > The experiments are conducted in an autoregressive manner, as correctly mentioned by the reviewer.
> >
> > **C10. The motivation for offline prediction synthesis discusses improved parameter efficiency, but parameter efficiency is not discussed in the rest of the paper. Why is modularity desirable in this setting?**
> >
> > Our intention was not to claim that modularity is desirable in our setting but rather to mention its importance in the referred paper. We apologize for any confusion caused and will make sure to clarify this point accordingly.
> >
> > **C11. Any intuition as to why MoAT_time is the best performer on the Fuel dataset?**
> >
> > It is important to note that we fine-tuned each variant of MoAT, including MoAT_time, within the same hyperparameter search space. In the case of the Fuel dataset, text behaves somewhat like noise and does not contribute to performance improvement. MoAT is a more universally applicable configuration across all datasets, providing a more generalized approach.
> >
> > **C12. The ablations in Table 2 suggest that the augmentation are not really helpful, considering the relative improvement compared to MoAT_time and MoAT_text.**
> >
> > The comparison between MoAT, MoAT_time, and MoAT_text aims to highlight the effectiveness of leveraging multiple modalities for time series forecasting. Conversely, MoAT_sample and MoAT_feature are intended to demonstrate how employing both augmentation methods using these multi-modalities enhances performance. Therefore, these different variants are designed to confirm distinct aspects rather than competing with each other.
> >
> > **C13. Ridge regression includes a loss penalty. What weight did you choose for this?**
> >
> > The weight for the regularization term in ridge regression is selected based on its performance evaluated on the validation set.
> >
> > **C14. Why are the prediction lengths so short?**
> >
> > We intentionally opted for shorter forecasting lengths due to the relatively smaller size of the datasets under consideration. A longer forecasting length would unavoidably reduce the number of available training samples, potentially causing performance degradation.

---

### Official Review · Reviewer_uJCG · 2023-11-06

**Soundness:** 2 fair
**Presentation:** 2 fair
**Contribution:** 2 fair
**Rating:** 5
**Confidence:** 4

**Summary:**

This paper focuses on multi-modal time series forecasting, particularly the text data augmented time series. It includes three main components, i.e., patch-wise embedding, multi-modal augmented encoder, and a trend-seasonal decomposition.
The experimental evaluation is through several financial datasets.

**Strengths:**

1. This paper focuses on an interesting applied problem, i.e., exploring the integration of text data into time series to enhance forecasting.
This problem is not new and widely studied in quantitative finance, data mining, etc.

2.  The experimental evaluation is conducted on several real financial price data across different markets. The authors augment the time series by collecting real news from news data providers. These datasets, if open-sourced, would be very helpful for the community.

**Weaknesses:**

1. This paper is mostly applied and combines several existing techniques, e.g., patch-wise embedding, and pattern decomposition.
The authors are expected to better position this work by clarifying the technical novelty, contribution, or new insights.

2. The evaluation is mostly on financial datasets. But for finance, the error metric MSE is not the main interest, since in the real world the prediction is to serve downstream tasks, e.g., portfolio construction, risk management, etc, and practically MSE does not directly translate to the improvement for downstream tasks. It would be better to show the prediction by the proposed method can facilitate an example downstream task. e.g., portfolio construction is commonly used.

**Questions:**

1. On page 5, the part "multi-modal augmented encoder" is essentially a combination of time series and text modalities, and what does the "augmented" refer to?

2. On page 6, in the part "joint trend-seasonal decomposition", trend-seasonal decomposition is reasonable for time series because of the underlying generative process. But, for text data, especially the embedding of news text, applying trend-seasonal decomposition is not intuitively understandable. News content is highly dependent on real-world happenings where the trend-seasonal decomposition is not necessarily existent.

3. In Table 1,  MSE are mostly low-magnitude values, while the real price of the finance assets of the data used in the experiment differs greatly in magnitudes. Is this due to some data standardization? If so, it is better to report the errors in the original scale of data, because the prediction on the standardized domain would be less useful for downstream tasks in many cases.

---

> ### Author Response · Authors · 2023-11-19
> **Dear Reviewer uJCG: Thank you for the review.**
>
> We appreciate your time and efforts in providing insightful comments. Below, we tried our best to address your concerns.

---

> > ### Author Response · Authors · 2023-11-19
> > **Responses to the weakness**
> >
> > **C1. This paper is mostly applied and combines several existing techniques, e.g., patch-wise embedding and pattern decomposition. The authors are expected to better position this work by clarifying the technical novelty, contribution, or new insights.**
> >
> > We would like to emphasize the technical contributions and novel insights our work brings to both the multi-modal and time-series domains as follows:
> >
> > - We have reformulated existing augmentation techniques within the multi-modal domain, categorizing them into sample-wise (modality-specific) and feature-wise (modality-fused) augmentations. This configuration of multi-modal augmentation provides novel and structured viewpoints on how multiple modalities can be effectively fused.
> > - Surprisingly, despite its potential, there was a lack of benchmark datasets incorporating both time series and text (or other modalities). Thus, we gathered six real-world datasets, and we plan to make them publicly accessible. We strongly believe these datasets will serve as valuable resources for future researchers in this field.
> > - We proposed MoAT, a time-series forecasting method that leverages multi-modal information. Specifically, MoAT consists of a multi-modal augmented encoder, which benefits from two distinct augmentations applied to both time series and text inputs. Note that most existing multi-modal approaches have typically utilized only one of these augmentation methods.
> > - As the reviewer commented, patching and decomposition are existing techniques in time series forecasting. However, we would like to emphasize that, to the best of our knowledge, no prior work has combined both patching and decomposition for time series forecasting. Our approach, integrating these techniques, has notably improved accuracy.
> >
> > In summary, we would like to emphasize that our work makes substantial contributions to (1) redefining methodologies, (2) introducing new datasets, and (3) presenting technically novel methods. As the reviewer suggested, we will emphasize these aspects in the revised version.
> >
> >
> > **C2. The evaluation is mostly on financial datasets. But for finance, the error metric MSE is not the main interest, since in the real-world, the prediction is to serve downstream tasks, e.g., portfolio construction, risk management, etc, and practically MSE does not directly translate to the improvement for downstream tasks.**
> >
> > The reviewer’s perspective is interesting, and we recognize the relevance of various downstream tasks tailored specifically for financial datasets. Nevertheless, we would like to emphasize that MoAT is a versatile time series forecasting framework that leverages multi-modal information. Thus, our evaluation primarily focused on measuring performance through MSE, which is a conventional metric used to assess the accuracy of the model.
> >
> >
> > **C3. On page 5, the part “multi-modal augmented encoder” is essentially a combination of time series and text modalities, and what does the “augmented” refer to?**
> >
> > In Section 2, we restructured two orthogonal approaches for augmenting multi-modal data. Figure 3 (a) illustrates the multi-modal encoder in MoAT, which utilizes both augmentation methods. This integration allows MoAT to leverage both modality-specific and modality-fused information extractable from time series and text data.
> >
> > **C4. For text data, especially the embedding of news text, applying trend-seasonal decomposition is not intuitively understandable.**
> >
> > We acknowledge the reviewer's comment and modified the attention pooling function for texts by using time series patches as query vectors during the attention computation for text aggregation. For more details, please refer to the common response above.
> >
> > **C5. In Table 1, MSE are mostly low-magnitude values, while the real price of the finance assets of the data used in the experiment differs greatly in magnitudes. It is better to report the errors in the original scale of the data, because the prediction on the standardized domain would be less useful for downstream tasks in many cases.**
> >
> > In line with existing practices in time-series forecasting research, we standardized the entire time series dataset using the training set and conducted training and evaluation accordingly. Although the standardized values may vary in magnitudes, it is important to note that the relative rankings among different methods remain consistent. In addition, it is straightforward to revert to the original scales through de-standardization using the mean and standard deviation obtained for standardization.

---

### Author Response · Authors · 2023-11-19
**Thank you for the reviews**

We deeply appreciate the valuable time and effort invested by the reviewers in thoroughly assessing our work. Their insightful feedback and comments have been instrumental in shaping and improving our research.

---

### Author Response · Authors · 2023-11-19
**Text Decomposition (Common Response)**

We acknowledge the feedback from the reviewers concerning the trend-seasonal decomposition of text data. In response to this feedback, we have made modifications to MoAT to enhance the clarity and effectiveness of text decomposition. Specifically, we modified the text aggregation function as:

$\text{Softmax}(\text{tanh}(d_{ta:tb}W + b) p_{ta:tb})d_{ta:tb}$

where $p_{ta:tb}$ is the time series patch embedding spanning from time ta to tb. Instead of utilizing a global query vector, we now incorporate the corresponding time series patch. This alteration allows for text aggregation based on the semantics of the time series. Notably, as the time series patch varies between trend and seasonal time series, texts are aggregated differently based on these distinctions. Our experimental results after the modification are as follows:

|            | Fuel  | Metal | Bitcoin | Stock-Index | Covid | Stock-Tech |
|------------|-------|-------|---------|-------------|-------|------------|
| Previous   | 0.0816| 0.0257| 0.0494  | 0.8134      | 0.1727| 0.1176     |
| New        | 0.0818| 0.0255| 0.0496  | 0.8131      | 0.1734| 0.1176     |

Our experimental results indicate that the performance following this modification shows marginal changes from the attention pooling function. The ranks in Table 1 remain the same.

---

### Comment · Area_Chair_4gnK · 2023-11-23
**From AC at the end of rebuttal: Reviewer response required**

Dear Reviewers,

Thanks for your time and commitment to the ICLR 2024 review process.

As we approach the conclusion of the author-reviewer discussion period (Wednesday, Nov 22nd, AOE), I kindly urge those who haven't engaged with the authors' dedicated rebuttal to please take a moment to review their response and share your feedback, regardless of whether it alters your opinion of the paper.

Your feedback is essential to a thorough assessment of the submission.

Best regards,

AC

---

### Meta-Review · Area_Chair_4gnK · 2023-12-10

**Metareview:**

This paper studies a new application of time series forecasting by augmenting with text data. Paper received negative reviews initially, while the rebuttal did not convince the reviewers sufficiently. In a nutshell, the introduction of the new dataset is useful, while it is well motivated from an application perspective to include other modalities to enhance time series forecasting tasks. However, the concerns still held by the reviewers after rebuttal are well grounded. The paper is insufficient in technical contribution and empirical insights. The benefit from the text modality is not pronounced. The performance benefit mainly comes from combining well-established methods such as patchify and decomposition.

**Justification For Why Not Higher Score:**

After rebuttal, still unanimous rejection. Main concerns in the lack of novelty and technical contributions.

**Justification For Why Not Lower Score:**

N/A

---

### Decision · Program_Chairs · 2024-01-16

Reject